# ROBUST LEARNING OF DIFFUSION MODELS WITH EXTREMELY NOISY CONDITIONS

## ABSTRACT

Conditional diffusion models have the generative controllability by incorporating external conditions. However, their performance significantly degrades with noisy conditions, such as corrupted labels in the image generation or unreliable observations or states in the control policy generation. This paper introduces a robust learning framework to address extremely noisy conditions in conditional diffusion models. We empirically demonstrate that existing noise-robust methods fail when the noise level is high. To overcome this, we propose learning pseudo conditions as surrogates for clean conditions and refining pseudo ones progressively via the technique of temporal ensembling. Additionally, we develop a Reverse-time Diffusion Condition (RDC) technique, which diffuses pseudo conditions to reinforce the *memorization effect* and further facilitate the refinement of the pseudo conditions. Experimentally, our approach achieves state-of-the-art performance across a range of noise levels on both class-conditional image generation and visuomotor policy generation tasks.

## 1 INTRODUCTION

Diffusion models (Ho et al., 2020; Song et al., 2021b; Karras et al., 2022; Chi et al., 2023) have significantly enhanced the generative tasks with success in multiple areas such as image generations (Kim et al., 2023; Singh & Raza, 2021; Song et al., 2021a; Karras et al., 2022; Rombach et al., 2022; Hatamizadeh et al., 2024), robotic control (Ma et al., 2024; Chi et al., 2023; Wang et al., 2023a; Li et al., 2024), and text generations (Li et al., 2022c; Gong et al., 2023; Wu et al., 2023; Gong et al., 2023) and with the continuous improvements on the generation efficiency (Song et al., 2021a; Shih et al., 2023). In particular, diffusion models gradually add random noise to the input $x_0$ through a forward stochastic process, resulting in increasingly noisy variants $x_t, t \in [0, T]$. A denoising model is then trained to reverse this process via score matching (Song et al., 2021b) to remove the random noise of $x_t$ towards $x_0$ over multiple time steps $t$.

In particular, conditional diffusion models have introduced the controllability (Dhariwal & Nichol, 2021; Wang et al., 2023b; Huang et al., 2023) by adding various types of conditions to guide the generation (Ni et al., 2023; Luo et al., 2023; Zhang et al., 2023; Cao et al., 2024). For example, in the label-condition image generation (Rombach et al., 2022; You et al., 2023; Ifriqi et al., 2024), the generated images should closely match the class labels, and conditions are labels. In the generation of visuomotor policy (Chi et al., 2023; Na et al., 2024), the diffusion model can generate coherent actions based on visual observations by robots, and conditions are the visual observations (images) and the current robotic states. Specifically, conditional diffusion models embed the conditions into the learning objective of the denoising score matching across all time steps $t$ and optimize the denoising model, given both the input $x_0$ and its corresponding conditions $y$.

However, the performing conditional diffusion models reply on high-quality conditions $y$ that are often noisy in practice due to unreliable data sourcing (Jiang et al., 2022). Noisy conditions decrease the controllability of the generation that could cause performance degradation (as shown in Figure 1 (b)) and even hazards. For example, in image generation, incorrect labels can make the diffusion models generate the misleading images (Dufour et al., 2024); In the visuomotor policy generation, unreliable visual observation can lead to hazardous behaviors in real-world deployment, such as collapsing the entire robot system in critical applications of autonomous driving (Kahn et al.,

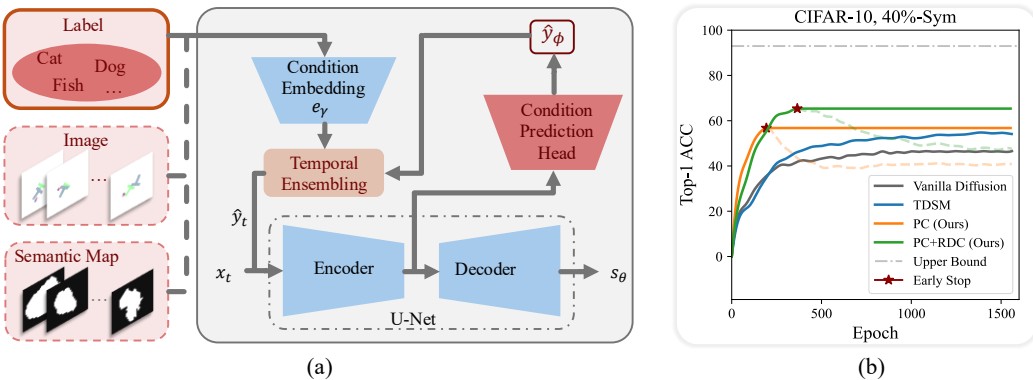

(a)                                                                 (b)

Figure 1: (a) Structures of our robust diffusion model: a lightweight prediction head that predicts pseudo conditions $\hat{y}$ is added at the output of the U-Net encoder of the diffusion model, and temporal ensembling is then adopted to update pseudo conditions. (b) Learning dynamics of conditional diffusion models on CIFAR-10 under $40\%$ symmetric noise. Y-axis: controllability (Top-1 ACC of generated images, 1k images/class, 10 classes); X-axis: training epochs. Generations are evaluated using a pretrained CIFAR-10 classifier (Top-1 ACC $92.89\%$, silver dash-dot line). We compare the pseudo condition (PC) in orange curve and PC with Reverse-time Diffusion Condition (RDC) in green curve, both with early stopping (star markers), against TDSM (Na et al., 2024) in blue and the vanilla conditional diffusion (Karras et al., 2022) in gray curve.

2017; Zhu et al., 2024; Getahun & Karimoddini, 2024), robotic manipulations (Kalashnikov et al., 2018), and surgeries (Murphy & Alambeigi, 2023; Fan et al., 2024).

There is a pioneering study on the label-noise diffusion models for the image generation (Na et al., 2024), and we find it fails on the high noise level. Na et al. (2024) estimated a noisy condition transition matrix (Yao et al., 2020) that can establish a linear relationship between clean and noisy labels. Then, the obtained noisy condition transition matrix was leveraged to weigh the output of denoising model. Highly noisy conditions often create *entangled clusters* in demonstrations (Zhang et al., 2024), reducing feature consistency and hinder the diffusion model from learning the discriminative representations in early training. This is evident by the *underfitting* phenomenon of Vanilla Diffusion (Karras et al., 2022) (gray curve) and TDSM (Na et al., 2024) (blue curve) in Figure 1(b).

Targeting the problem of noisy distributions $p_0(x|\tilde{y})$, this paper introduces a pseudo condition $\hat{y}$ as a surrogate of the clean condition $y$. Under the classifier-free guidance framework (Ho & Salimans, 2022), we construct a lightweight prediction head that predicts pseudo conditions $\hat{y}$ of originally noisy counterparts $\tilde{y}$ (as shown in the Figure 1 (a)). During the optimization, the pseudo condition $\hat{y}$ can gradually refine itself and replace the noisy $\tilde{y}$, attributed to the memorization effect of fitting clean conditions before the noisy counterparts (Patrini et al., 2017; Han et al., 2018). Specifically, we update $\hat{y}$ using the technique of temporal ensembling (Laine & Aila, 2017) so that it gradually refines itself. Early stopping is then empirically applied to prevent $\hat{y}$ from overfitting to $\tilde{y}$, as shown by the translucent dashed orange/green curves in Figure 1 (b).

Furthermore, we propose a technique of Reverse-time Diffusion Condition (RDC) to transform the $\hat{y}$ to its diffusion process $\hat{y}_t$ ($t \in [0, T]$) that can further facilitate the refinement of conditions. Specifically, $\hat{y}_0$ represents a completely random condition, $\hat{y}_T$ represents the pseudo condition $\hat{y}$, and $\hat{y}_t$ ($t \in (0, T)$) represents $\hat{y}$ perturbed with a random noise at time $t$. Our robust learning objective becomes score matching between the input $(x_t, \hat{y}_t)$ and its target $(x_0, \hat{y}_T)$, and the U-Net model (Ronneberger et al., 2015) is then trained to fit this matching across various time steps $t \in [0, T]$. Specifically, RDC injects additional randomness into the already noisy conditions during the training and compels the optimization of the denoising model under these additional randomness that serves as a form of condition augmentation. In Figure 1 (b)that green line outperforms yellow line shows RDC can significantly enhance the memorization effect.

We have conducted experiments on two condition generation task: visuomotor policy generation (Chi et al., 2023) and image generation (Na et al., 2024). For visuomotor policy generation, we have conducted experiments on Push-T dataset (Florence et al., 2021) with the noisy conditions

when image observations are corrupted by camera distortion. For image generation, we have used both the CIFAR-10 and CIFAR-100 datasets (Krizhevsky et al., 2009) under two types of label-noise: symmetric noise (van Rooyen et al., 2015) and asymmetric noise (Patrini et al., 2017). All experiments corroborates our RDC-powered robust diffusion learning achieved the state-of-the-art (SOTA) performances across various levels of condition noises.

## 2 PRELIMINARY

### 2.1 SCORE MATCHING FOR DIFFUSION MODELS

Given a demonstration distribution $p_0(x)$, score matching (Hyvärinen, 2005) is to estimate the gradient of the log-density function without requiring explicit normalization of the probability distribution, i.e., $s_\theta(x) = \nabla_x \log p_0(x)$, where $x \in \mathbb{R}^d$, $\nabla_x \log p_t(x)$ is the score of $x$. Then score matching minimizes the expected squared difference between the estimated and score functions.

Song et al. (2021b) applied score matching for generation with transforming $x$ to a diffusion process $x_t$ with multiple score matching objectives at time $t \in [0, T]$. The distribution of $x$ is first perturbed through a forward stochastic differential equation (SDE):

$$\mathrm{d}x = f(x,t)\mathrm{d}t + g(t)\mathrm{d}\mathbf{w}, \tag{1}$$

where $f(\cdot, t) : \mathbb{R}^d \to \mathbb{R}^d$ is the drift coefficient determining the perturbation direction of demonstration $x$ over time $t$; $g(\cdot) : \mathbb{R} \to \mathbb{R}$ is the diffusion coefficient that controls the level of random noise applied to $x$ at the time $t$; $\mathbf{w}$ is a standard Wiener process(Gelbrich & Römisch, 1995). In the widely adopted EDM (Karras et al., 2022) implementation, $f(\cdot, t) = 0$ and $g(\cdot) = \sqrt{2t}$. This SDE gradually transforms the distribution of $x$ into a simple one (e.g., normal) as $t \to T$.

Then a reverse-time SDE is proposed to transform simple distribution back to the distribution of $x$:

$$\mathrm{d}x = \left[ f(x,t) - g^2(t)\nabla_x \log p_t(x) \right] \mathrm{d}t + g(t)\mathrm{d}\bar{\mathbf{w}}, \tag{2}$$

where $\bar{\mathbf{w}}$ denotes another standard Wiener process (Gelbrich & Römisch, 1995).

The score $\nabla_x \log p_t(x)$ is approximated using a U-Net $s_\theta(x_t, t) : \mathcal{X} \times \mathcal{T} \to \mathcal{X}$, where $\mathcal{X} \subseteq \mathbb{R}^d$ denote the demonstration space and $\mathcal{T} \subseteq \mathbb{N}$ is the space of time steps. We minimize the objective of the denoising score matching :

$$\mathbb{E}_{t \sim \mathcal{U}(0,T)} \left[ \lambda(t) \cdot \mathbb{E}_{x_t \sim p_t} \|s_\theta(x_t, t) - \nabla_{x_t} \log p_t(x_t)\|_2^2 \right], \tag{3}$$

where $\mathcal{U}(0, T)$ denotes the uniform distribution over $[0, T]$, $\lambda(t)$ is a weighting function, $p_0$ is the demonstration $x$ distribution, and $p_t(x_t)$ is the transition distribution defined by the forward SDE.

### 2.2 CLASSIFIER-FREE GUIDANCE (CFG)

CFG introduces the controllability into diffusion generation by modeling the conditional distribution $p_0(x \mid y)$ (Ho & Salimans, 2022). It allows conditional sampling without using an external classifier. Giving demonstration-condition pair $(x,y)$, a conditional U-Net model $s_\theta(x_t, t, y) : \mathcal{X} \times \mathcal{T} \times \mathcal{Y} \to \mathcal{X}$, where $\mathcal{Y} \subseteq \mathbb{R}^d$ is the condition space, is trained to predict the conditional score of the demonstration $x$ at time $t$ given condition $y$. Besides that, an unconditional model $s_\theta(x_t, t, \varnothing)$ is also trained by randomly dropping $y$ (i.e., setting as all-zero vector in the label condition) with a certain probability.

The conditional denoising score matching objective for the demonstration is $\operatorname{argmin}_\theta \mathcal{L}_{\mathrm{demo}}$, where

$$\mathcal{L}_{\mathrm{demo}} = \mathbb{E}_{t \sim \mathcal{U}(0,T)} \left[ \lambda(t) \cdot \mathbb{E}_{x_t \sim p_t, y \sim p(y)} \|s_\theta(x_t, t, y) - \nabla_{x_t} \log p_t(x_t|y)\|_2^2 \right]. \tag{4}$$

At sampling time, the conditional score prediction $\nabla_{x_t} \log p_{t|0}(x_t|x_0, y)$ is a weighted combination of conditional prediction and unconditional prediction:

$$s_{\mathrm{CFG}}(x_t, t, y) = s_\theta(x_t, t, \varnothing) + w \cdot (s_\theta(x_t, t, y) - s_\theta(x_t, t, \varnothing)), \tag{5}$$

where $w \geq 1$ is the guidance scale. Large $w$ increases the controllability of the model with respect to the condition $y$. For consistency, all symbols appearing throughout this paper are listed in Table 4.

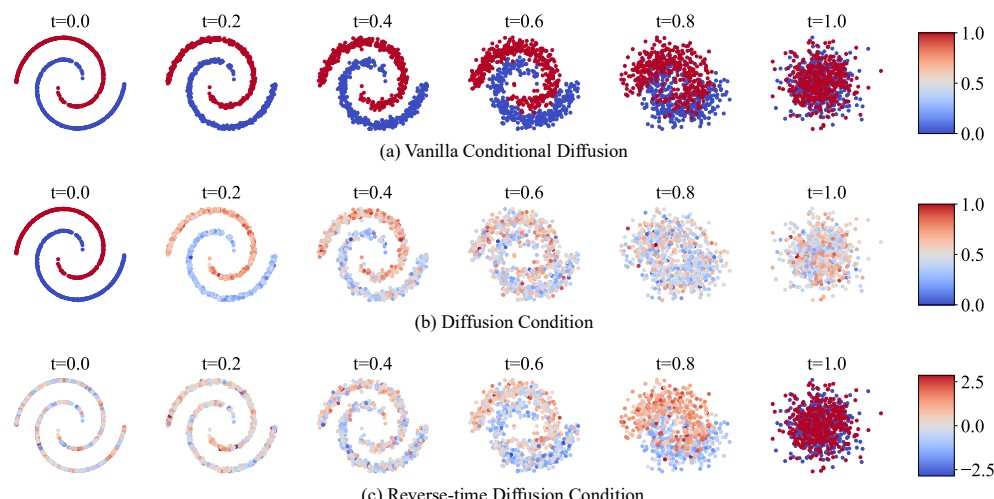

Figure 2: Visualization of three diffusion behaviors on the condition from $t=0$ to 1. (a) Standard conditional diffusion: $y_t$ remains fixed. (b) Forward-diffused condition: $y_t$ gradually becomes Gaussian. (c) Reverse-time diffusion condition (RDC): $y_t$ evolves back toward $y$.

## 3 METHOD

### 3.1 CONDITIONAL DIFFUSION LEARNING UNDER NOISY CONDITIONS

#### 3.1.1 PSEUDO CONDITION

The starting point of our method is the observation that, under noisy conditions, the conditional distribution $p_0(x|\tilde{y})$ becomes entangled, as each cluster of noisy condition $\tilde{y}$ corresponds to demonstration $x$ originating from multiple clusters of clean conditions $y$. Such entanglement undermines feature consistency and makes it difficult for the diffusion model to learn valuable representations during early training. Consequently, vanilla conditional diffusion suffers from poor controllability, as shown in Figure 1(b). This raises the central motivation of our method: can noisy conditions be corrected by explicitly breaking their internal entanglement?

Our method replace this noisy clustered distribution $p_0(x|\tilde{y})$ with $p_0(x|\hat{y})$, where $\hat{y}$ is the proposed pseudo condition refined during training. As shown in Eq. 5, $s_{\text{CFG}}(x_t, t, \tilde{y})$ is switched into $s_{\text{CFG}}(x_t, t, \hat{y})$. Firstly, for each demonstration $x_i$ in the dataset $\tilde{D}$, we construct and initialize $\hat{y}_i$ as an all-zero vector that has the same shape as $\tilde{y}_i$. Assigning the same initial pseudo-condition to all demonstrations effectively disrupts the entangled clusters in the original dataset. Then we update the pseudo condition $\hat{y}$ to approximate the clean condition $y$. As shown in the Figure 1 (a), we add a lightweight prediction head $q_\phi(x_t, t, \hat{y})$ at the output of the diffusion model U-Net encoder to obtain the refinement $\hat{y}_\phi = q_\phi(x_t, t, \hat{y})$ of the pseudo condition $\hat{y}$. Significantly, this additional prediction head incurs negligible additional computational overhead thanks to the parameter sharing in the U-Net encoder. We then optimize the refinement of the pseudo condition with the following learning objective $\operatorname{argmin}_\phi \mathcal{L}_{\text{cond}}$:

$$\mathcal{L}_{\text{cond}} = \|\hat{y}_\phi - \tilde{y}\|^2. \tag{6}$$

In the early training stage, the memorization effect (Liu et al., 2020) enables demonstrations $x$ with similar features are more likely to be clustered together, so that we can obtain a better pseudo condition $\hat{y}$ compared with the original noisy condition $\tilde{y}$ by updating $\hat{y}$ in the early training stage using the temporal ensembling(Laine & Aila, 2017):

$$\hat{y} = \alpha\hat{y} + (1-\alpha)\hat{y}_\phi, \tag{7}$$

where $0 \leq \alpha \leq 1$ is the momentum. Finally, we employ early stopping with an empirically determined criterion to prevent the clustered distribution from overfitting to the noisy distribution $p_0(x|\tilde{y})$. Please refer to Appendix 5 for more details.

### 3.1.2 REVERSE-TIME DIFFUSION CONDITION

Inspired by Kingma & Gao (2023), which view diffusion training as data augmentation through Gaussian noise of varying scales, we extend this idea by perturbing pseudo conditions within the diffusion process to further enhance their learning. As shown in Fig. 2, diffusing the condition along the demonstration direction (Fig. 2(a)) forces the model at $t = T$ to infer both demonstration and condition information from the fully corrupted $(x_T, y_T)$, leading to large errors and unstable training. In contrast, our RDC (Fig. 2(b)) keeps $y_T = y$ while $x_T$ becomes Gaussian, substantially reducing inference difficulty and stabilizing training.

Motivated by the need to stabilize training under noisy conditions, we transform pseudo condition $\hat{y}$ into a reverse-time diffusion process. First, we define the two critical state of $\hat{y}$ in the process, when $t = 0$, $\hat{y}_0 \sim \mathcal{N}(\mu, \sigma)$, ($\mu, \sigma$ are the specified mean and standard deviation), and when $t = T$, $\hat{y}_T = \hat{y}$. Then the reverse-time diffusion condition $\hat{y}_t$ is defined as:

$$\begin{cases} \text{Forward SDE:} \ \ \mathrm{d}\hat{y} = \dfrac{-f(\hat{y}, t)}{g(t)}\mathrm{d}t + \dfrac{1}{g(t)}\mathrm{d}\mathbf{w}, \\[3mm] \text{Reverse SDE:} \ \ \mathrm{d}\hat{y} = \left( \dfrac{-f(\hat{y}, t)}{g(t)} - \dfrac{1}{g(t)^2}\nabla_{\hat{y}} \log p_t(\hat{y}) \right)\mathrm{d}t + \dfrac{1}{g(t)}\mathrm{d}\bar{\mathbf{w}}, \end{cases} \tag{8}$$

where $f(\cdot, t)$ and $g(t)$ have the same definition as Eq. 1, $w$ and $\bar{w}$ are two standard Wiener process (Gelbrich & Römisch, 1995) as well, following the standard formulation of the SDEs, detailed derivation of RDC SDEs could be found at Appendix 4.1.

Correspondingly, we transform the prediction target of the condition prediction head from the refinement of the $\hat{y}_\phi$ into the condition score matching objective $s_\phi(x_t, t, \hat{y}_t)$ , to learning the condition score function $\nabla_{\hat{y}} \log p_t(\hat{y})$ in the forward SDE. Then the prediction of pseudo condition $\hat{y}_\phi$ can be computed in the following integral form:

$$\hat{y}_\phi = \hat{y}_0 - \int_0^T \frac{s_\phi(\hat{y}_t, t)}{2t} \, \mathrm{d}t, \tag{9}$$

where the derivation for the integral form of $\hat{y}_\phi$ is given in the Appendix 4.2.

Next, we consider the optimization objective. For the demonstrations $x$, the objective $\mathcal{L}_{\text{demo}}$ remains unchanged as defined in Eq. 4. For the pseudo-condition $\hat{y}$, instead of following the optimization of $x$ to estimate the condition score function $s_\phi(x_t, t, \hat{y}t)$, we directly optimize the denoised condition $\hat{y}_\phi$ in the Eq. 9 for the pseudo-condition update in the Eq. 7. Specifically, $\hat{y}_\phi$ is estimated using numerical methods (Gelbrich & Römisch, 1995), and the overall learning objective combines the conditional denoising score matching objective and the reverse-time diffusion condition learning objective $\text{argmin}_{\theta,\phi}\mathcal{L}$:

$$\mathcal{L} = \text{argmin}_\theta \mathcal{L}_{\text{demo}} + \mathbb{E}_{t \sim \mathcal{U}[0,T]} \|\hat{y}_\phi - \tilde{y}\|^2 \tag{10}$$

Note that $\hat{y}_\phi$ is obtained by integrating the learned condition score $s_\phi$ via the SDE-equivalent ODE (Eq. 9). Thus, the RDC loss remains a form of score matching rather than simple regression.

According to Kingma & Gao (2023), the RDC objective $\mathcal{L}_{\text{RDC}}$ is approximately equivalent to minimizing the standard conditional negative log-likelihood plus an explicit regularization term:

$$\mathcal{L}_{\text{RDC}} \approx \arg\min_\theta \left\{ -\mathbb{E}_{t,x_t} \left[ \log p_\theta(x_t | \hat{y}) \right] - \mathbb{E}_{t,x_t} \left[ \frac{1}{2g(t)^2}\Delta t \cdot \text{Tr}(\nabla_{\hat{y}}^2 \log p_\theta(x_t | \hat{y})) \right] \right\} \tag{11}$$

where the second term is the explicit time-annealed hessian regularization, with a strength inversely proportional to the square of the original diffusion coefficient. Derivation can be found in Appendix 4.3.

### 3.2 ALGORITHM

In this section, we extend our method to more challenging conditions such as images, where the condition embedding is implemented by a condition encoder $e_\gamma(\cdot)$ that maps noisy images to $\tilde{y}$. The main challenge is the joint optimization of the encoder $e_\gamma$ and the diffusion U-Net $s_\theta$.

The following adjustments are made to support the additional conditional encoder, with a stepwise description provided in Algorithm 1. Consistent with Section 3.1.1, we firstly construct the pseudo-condition by initializing $\hat{y}$ with the output embedding $\tilde{y}$ of the condition encoder, since uninformative embeddings at the start of training can effectively disrupt the entangled clusters present in the original dataset. A lightweight prediction head is also added to the encoder output of the U-Net to facilitate subsequent updates of the pseudo-condition. Then the pseudo-condition is converted into a RDC. we set the critical state when $t = T$ as $\hat{y}_T = \tilde{y}$, and refine the pseudo condition $\hat{y}$ with the same operations as Section 3.1 outlined. After early stopping, the condition encoder $e_\gamma$ continues to refine its output embedding $\tilde{y}$ toward the learned pseudo-condition $\hat{y}$, maintaining alignment between the encoder representation and the updated condition. The learning objective of the condition encoder during the later training stage is $\mathrm{argmin}\gamma\mathcal{L}_{\mathrm{enc}}$, where $\mathcal{L}_{\mathrm{enc}} = \|\tilde{y} - \hat{y}\|^2$.

---

**Algorithm 1** Robust Conditional Diffusion Learning with RDC

**Input:** Dataset $\tilde{\mathcal{D}} = \{(x, \tilde{y})\}$; condition encoder $e_\gamma$ (optional); score network $s_\theta$; predictor $q_\phi$.
**Output:** Trained $\theta$, and $\phi$; (if $e_\gamma$ used) trained $\gamma$.

1  **if** *using encoder* **then**
2      initialize pseudo condition $\hat{y}$ with the output $\tilde{y}$ of the condition encoder $e_\gamma$;
3      **else** initialize pseudo condition $\hat{y}$ with 0;
4  **end**
5  **while** *not early stopping* **do**
6      sample $t \sim \mathcal{U}(0, T)$; $x_t \sim p_t(x_t)$; $y_t \sim p_t(y_t)$;
7      update pseudo condition $\hat{y}_\phi \leftarrow q_\phi(x_t, t, y_t)$; $y_t \leftarrow (1-\alpha)y_t + \alpha\hat{y}_\phi$;
8      update $\theta, \phi$ with $\mathcal{L}_{\mathrm{cond}}$ (6) and $\mathcal{L}_{\mathrm{demo}}$ (4);
9  **end**
10  **if** *using encoder* **then**
11      update $\gamma$ with $\mathcal{L}_{\mathrm{enc}} = \|\tilde{y} - \hat{y}\|^2$;
12      replace $\hat{y}_t$ with $\tilde{y}$;
13  **end**
14  **else** replace $\hat{y}_t$ with $\hat{y}$;
15  Update $\theta$ with $\mathcal{L}_{\mathrm{demo}}$ (Eq. 4);

---

## 4 EXPERIMENT

We evaluated our method on several tasks, including 2-D data generation (Section 4.1), image generation conditioned on labels (experiments on symmetric noise in Section 4.2, and experiments on asymmetric noise in Appendix 5.5), visuomotor policy generation conditioned on images (Section 4.3), and image generation conditioned on semantic maps (Section 5.6). Our approach achieves SOTA performance in most of tasks.

### 4.1 2-D TOY CASE

Figure 3 shows a toy example of 2-D synthetic data generation, which is to visualize the conditional generation performance of our method. The synthetic dataset consists of four classes with 2k 2-D synthetic data for each class. We then compare our method with the vanilla diffusion model EDM (Karras et al., 2022), and the SOTA robust conditional learning diffusion model, TDSM (Na et al., 2024), under 20%, 40%, 60%, and 80% symmetric noise. We use the $x$ and $y$ axes in all these sub-figures in the Figure 3 to represent the two feature dimensions of the 2-D data. See more experimental setup in the Appendix 5.

In 2-D generation tasks, the Mean Absolute Error (MAE) quantifies how closely the generated data match the corresponding training data (noise level 0%) across both coordinate axes. A smaller MAE indicates closer alignment between the generated data and the clean data.

The noise level increases from left to right in each row of Figure 3, and we can see that our method surpass TDSM (Na et al., 2024) at all noise level from 20% to 80%. At 80% noise, TDSM shows

no improvement over the baseline, while our method still reduces MAE by about 0.5, indicating that TDSM method based on noisy condition transition matrix estimation fails under high noise.

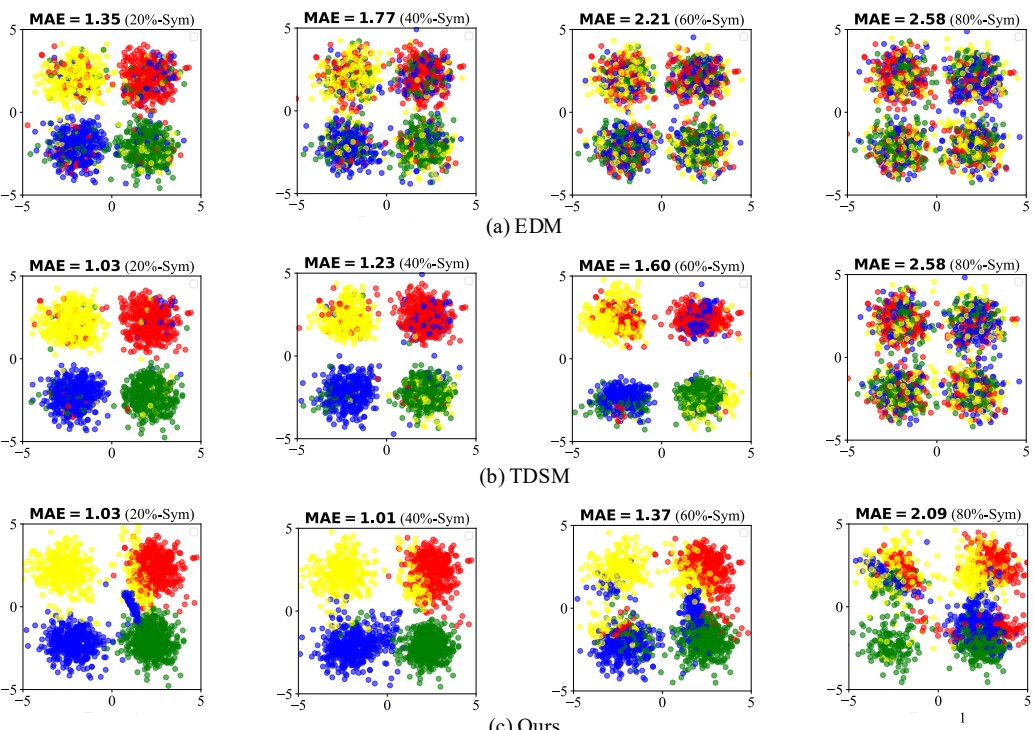

Figure 3: 2-D Toy Case. Comparison of robust condition learning methods using 2-D synthetic data in four class (red for class 1, blue for class 2, green for class 3, yellow for class 4). From left to right, three methods are trained and generated on 2-D data with symmetric noise, and the noise level equals 20%, 40%, 60%, and 80%.

## 4.2 IMAGE GENERATION CONDITIONED ON LABELS

We conducted class-label image generation experiments on the noisy variants of CIFAR-10 and CIFAR-100 datasets, where the class-label is condition, and the image is the demonstration. In this task, we experimented with two types of label noise separately. For symmetric noise, class labels were uniformly mislabeled as any other class, and experiments were conducted at noise levels $\eta = 20\%$, $40\%$, $60\%$, and $80\%$. For asymmetric noise, class labels were flipped to similar classes, with experiments conducted at noise levels $\eta = 20\%$ and $40\%$. Metrics contain Fréchet Inception Distance (FID) (Heusel et al., 2017), Inception Score (IS) (Salimans et al., 2016), Density, Coverage (Naeem et al., 2020) and Class-Wise (CW) metrics (Chao et al., 2022) (evaluate the average value of the above metric separately for each class). The experimental settings almost the same with Karras et al. (2022). See more details of hyperparameter selection such as early stopping rule in Appendix 5.

Table 1 presents the experimental results under symmetric noise conditions. We compared our method with two models: the vanilla diffusion model EDM (Karras et al., 2022) and the SOTA robust image generation model TDSM (Na et al., 2024) on the CIFAR-10 and CIFAR-100 datasets. The results show that our method significantly outperforms the current SOTA model across all noise levels. Specifically, our method achieves more than 10% better performance than TDSM (Na et al., 2024) on all class-wise metrics. The advantage of our method remains notable at high noise levels. When the noise level exceeds 60%, our method maintains far better performance compared to other methods. Notably, TDSM (Na et al., 2024) often collapses during training on CIFAR-100 with symmetric noise levels above 60%, so we omit their results for this setting.

Table 1: Conditional Generation Performance Comparison with EDM and TDSM on the CIFAR-10 and CIFAR-100 datasets under Symmetric Noise.

| Dataset | Noise Level | Method | FID ($\downarrow$) | IS ($\uparrow$) | Density ($\uparrow$) | Coverage ($\uparrow$) | CW-FID ($\downarrow$) | CW-Density ($\uparrow$) | CW-Coverage ($\uparrow$) |
|---|---|---|---|---|---|---|---|---|---|
| | 0% | EDM | 1.92 | 10.03 | 103.08 | 81.90 | 10.23 | 102.63 | 81.57 |
| | | EDM | **2.00** | 9.91 | 100.03 | 81.13 | 16.21 | 88.45 | 77.80 |
| | 20% | TDSM | 2.06 | 9.97 | 106.13 | 81.89 | 12.16 | 99.52 | 80.29 |
| | | **Ours** | 2.02 | **10.05** | **107.90** | **94.28** | **10.24** | **106.24** | **93.84** |
| | | EDM | **2.07** | 9.83 | 100.94 | 80.93 | 30.45 | 73.02 | 71.63 |
| | 40% | TDSM | 2.43 | 9.96 | **111.63** | 82.03 | 15.92 | 97.80 | 78.65 |
| CIFAR-10 | | **Ours** | 2.17 | **10.04** | 105.88 | **93.80** | **10.64** | **102.96** | **93.18** |
| | | EDM | 3.67 | 9.70 | 99.14 | 83.99 | 51.69 | 53.47 | 74.12 |
| | 60% | TDSM | **3.22** | 9.67 | **100.19** | 86.74 | 41.56 | 62.63 | 80.48 |
| | | **Ours** | 3.23 | **9.68** | 99.33 | **86.84** | **33.53** | **68.00** | **81.56** |
| | | EDM | 5.84 | 9.45 | 99.36 | 61.73 | 79.42 | 38.40 | 51.18 |
| | 80% | TDSM | 5.47 | **9.70** | 102.52 | 61.96 | 78.98 | 39.91 | 53.80 |
| | | **Ours** | **4.25** | 9.58 | **103.53** | **78.73** | **68.39** | 47.35 | **56.70** |
| | 0% | EDM | 2.51 | 12.80 | 87.98 | 77.63 | 66.97 | 82.58 | 75.78 |
| | | EDM | **2.96** | 12.28 | 83.01 | 75.02 | 79.91 | 66.47 | 70.11 |
| | 20% | TDSM | 4.26 | 12.29 | 85.66 | 74.90 | 78.71 | 70.62 | 70.77 |
| | | **Ours** | 3.18 | **12.95** | **98.32** | **93.52** | **71.57** | **90.49** | **91.51** |
| | | EDM | **3.36** | 11.86 | 81.70 | 73.92 | 100.04 | 49.77 | 60.64 |
| | 40% | TDSM | 6.85 | 12.07 | **88.45** | 72.12 | 93.24 | 60.60 | 63.89 |
| CIFAR-100 | | **Ours** | 4.60 | **12.73** | 84.75 | **89.25** | **76.56** | **75.74** | **87.90** |
| | | EDM | 7.07 | **12.54** | **93.55** | 83.53 | 117.75 | 42.92 | 74.37 |
| | 60% | TDSM | - | - | - | - | - | - | - |
| | | **Ours** | **5.57** | 12.03 | 91.89 | **87.45** | 104.34 | 67.39 | 84.03 |
| | | EDM | **11.13** | **12.66** | **92.09** | 71.53 | 146.97 | 25.02 | 52.57 |
| | 80% | TDSM | - | - | - | - | - | - | - |
| | | **Ours** | 11.50 | 10.94 | 83.08 | **73.03** | **133.09** | **35.64** | **59.98** |

For more experimental results on the asymmetric noise setting on both CIFAR-10 and CIFAR-100 datasets. Please refer to Appendix 5 for more details.

### 4.3 VISUOMOTOR POLICY GENERATION CONDITIONED ON IMAGES

We conducted visuomotor policy generation experiments conditioned on images from the noisy Push-T dataset (Florence et al., 2021). The task is to push a gray T-shaped block from a random position to a green target using image observations. To simulate condition noise, we apply two camera distortions: radial, which magnifies the image center, and tangential, which stretches regions due to camera misalignment. Distortions are applied with probability $\eta$ within predefined intensity thresholds. Policies are evaluated via the Target Area Coverage (TAC) metric (Chi et al., 2023), measuring the IoU between the block and target. Results are averaged over three training seeds and 500 random environment initializations. More implementation details can be found in Appendix 5.

We compare our method with Diffusion Policy (DP) (Chi et al., 2023) as the baseline, and further introduce the SOTA image distortion correction method MOWA (Liao et al., 2025) into the image pre-processing stage of DP (Chi et al., 2023), which is called "MOWA + DP". The latter compari-

Table 2: TAC Comparison with DP and MOWA+DP on the Push-T datasets under camera distortions.

| Method | Noise Level | | | |
|---|---|---|---|---|
| | 20% | 40% | 60% | 80% |
| DP | 76.64±1.67 | 73.02±2.53 | 68.35±5.00 | 68.46±3.31 |
| MOWA + DP | 77.80±**0.58** | 73.32±3.21 | 71.89±3.42 | 71.67±**2.09** |
| **Ours** | **80.26**±1.07 | **73.44**±1.42 | **72.74**±1.21 | **71.78**±3.24 |

son is designed to examine how far simply applying advanced denoising can achieve robust policy learning from noisy visual observations.

As shown in Table 2, our method demonstrates significant improvements over the DP (Chi et al., 2023) and MOWA + DP method across noise levels from $20\%$ to $80\%$ on Push-T dataset (Florence et al., 2021). Moreover, compared to the two-stage paradigm "MOWA + DP", our end-to-end approach is more streamlined and computationally efficient.

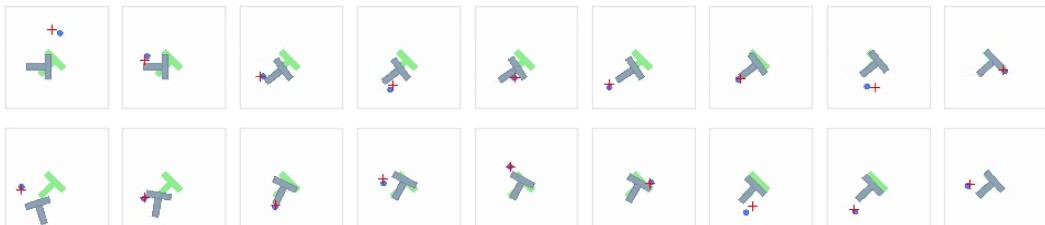

Figure 4: Results on Push-T with $80\%$ camera distortion. Each row shows one policy with nine key frames sampled at equal intervals.

To better illustrate our task, Figure 4 shows visuomotor policies trained on the Push-T dataset with $80\%$ camera distortion noise. Each row shows one generated policy from a random initial state, with nine equally spaced frames representing the main steps. These results show that our model can still learn and perform complex tasks under extreme observation noise, highlighting its robustness.

### 4.4 ABLATION STUDY

Table 3: Ablation results comparing vanilla diffusion (EDM) with(out) our PC and RDC modules on CIFAR-10 with 40% symmetric noise.

| Method | CW-FID ($\downarrow$) | CW-Density ($\uparrow$) | CW-Coverage ($\uparrow$) |
|---|---|---|---|
| EDM | 30.45 | 73.02 | 71.63 |
| Ours(PC) | 37.09 | 54.04 | 64.38 |
| **Ours(PC+RDC)** | **10.64** | **102.96** | **93.18** |

**Ablation Study on the Effectiveness of pseudo condition and RDC Mechanisms.** To evaluate the contribution of the proposed pseudo condition and RDC method, we conducted a set of ablation experiments to compare three configurations: (1) the vanilla diffusion model, (2) the vanilla diffusion model with the proposed pseudo condition, and (3) the vanilla diffusion model with both the proposed pseudo condition and RDC mechanism. We share the same experimental setting across all three experiment, the results shown in the Table 3 demonstrate that inappropriate implementation of adding pseudo condition can even do harm to the vanilla diffusion model EDM (Karras et al., 2022), as shown by the orange curve in Figure 1 (b) during the later stages of training. After we add RDC on top of pseudo condition, we can see a notable conditional generation improvement compared with the vanilla diffusion model EDM (Karras et al., 2022). These findings validate the combined effectiveness of pseudo condition and RDC in improving the model's generation capabilities.

### CONCLUSION

This paper addresses the problem of extremely noisy conditions in conditional diffusion models. We propose to learn pseudo conditions as surrogates for clean conditions and refine them progressively via the technique of temporal ensembling. Moreover, We improve pseudo condition learning using the RDC technique, which enhances memorization and facilitates the refinement of pseudo conditions through the reverse-time diffusion process. Experiments on both class-conditional image generation and visuomotor policy generation tasks shows that our method achieves SOTA performance across a range of noise levels.

## REPRODUCIBILITY STATEMENT

We are committed to ensuring the reproducibility of our results. To this end, we outline the following key components we provide for replication:

- **Datasets:** Appendix 5.1 provides the detailed processing steps used to synthesize noisy condition from the publicly available datasets. Besides that, all datasets used in our experiments will be released after the blind-review phase.
- **Experimental Setup:** All hyperparameters, such as learning rate, batch size, optimizer type, and training schedule, are documented in Appendix 5.3.
- **Code:** The implementation of our method, including training/evaluation scripts and configuration files, will be released after the blind-review phase.

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

## APPENDIX

## 1 LLM USAGE STATEMENT

We confirm that LLMs were used only for language polishment of this manuscript. They did not contribute to the research ideation and/or writing to the extent.

## 2 RELATED WORK

### 2.1 CONDITION-NOISE LEARNING

Research on noisy-condition has been extensive, and prior works have approached this problem from multiple directions, including data-centric strategies that filter or correct noisy conditions, model-based approaches that explicitly characterize noise distributions, and optimization techniques that introduce regularization or noise-robust training objectives.

From the data perspective, sample selection or reweighting methods (Han et al., 2018; Jiang et al., 2018; Ren et al., 2018; Li et al., 2020; 2022a) mitigate the impact of condition noise by selecting training demonstrations with clean conditions or reducing the weights of noisy demonstrations. Empirical studies on the memorization effect (Malach & Shalev-Shwartz, 2017; Liu et al., 2020) show that DNNs tend to first learn simple and clean conditions before gradually overfitting to all noisy ones. Many methods exploit this phenomenon to filter out demonstrations with clean conditions that are easier to fit (Gui et al., 2021; Wei et al., 2022). However, determining the proper timing of early learning is challenging (Xia et al., 2021; Bai et al., 2021), and non-stop iterative selection or reweighting may accumulate errors.

To address this limitation, condition correction methods (Tanaka et al., 2018; Ma et al., 2018; Liu et al., 2020; Zheng et al., 2021) adopt a gentler approach that considers all demonstrations and iteratively refines noisy conditions with soft/hard network predictions. Many of these methods leverage semi-supervised learning and contrastive learning, treating noisy demonstrations as unconditional ones while using classification predictions as pseudo conditions (Yang et al., 2022; Fooladgar et al., 2024).

From the model perspective, methods based on the noisy condition transition matrix aim to learn the mapping between the distributions of clean and noisy conditions by estimating the noisy condition transition matrix (Goldberger & Ben-Reuven, 2017; Zhang et al., 2021; Li et al., 2022b; Kye et al., 2022). However, most of these methods follow the impractical assumption of instance-independent

noise (Xia et al., 2020; Kye et al., 2022), and they usually fail under extremely noisy conditions due to large estimation errors of the transition matrix (Yao et al., 2020).

From the optimization perspective, noise-robust regularization prevents model parameters from overfitting noisy conditions by adding regularization terms to the loss function (Ghosh et al., 2017; Zhang & Sabuncu, 2018; Zhou et al., 2021; Ye et al., 2025). Beyond typical regularization techniques such as dropout and data augmentation, more advanced approaches have been proposed to handle higher levels of noise (Tanno et al., 2019; Xia et al., 2019). The main challenge of this line of work lies in the delicate design of optimization strategies (Huang et al., 2019; Ma et al., 2020).

## 2.2 CONDITION-NOISE LEARNING IN GENERATION

In generation tasks, conditional information or priors (such as class labels or text descriptions) are often introduced to enable controllable generation (Odena et al., 2017; Miyato & Koyama, 2018), which also brings challenges related to condition noise. Mismatched demonstration-condition pairs can degrade the quality of conditional generation, motivating research on robust methods.

One line of work focuses on conditional GANs. RCGAN (Thekumparampil et al., 2018) considers both known (RCGAN) and unknown (RCGAN-U) noise distributions. RCGAN introduces a known noise channel in the generator's output conditions and combines it with a projection discriminator to enhance adaptability to noise. RCGAN-U jointly optimizes the generator and a dynamically learned noise model (confusion matrix) to approximate the true conditional distribution under unknown noise. Similarly, Kaneko et al. (2019) proposed AC-GAN (with an auxiliary classifier) and cGAN (without any auxiliary classifier), incorporating a noise transition matrix into the classifiers and discriminators and adding noise-robust regularization to the loss functions. While effective, these methods struggle under high noise levels, and computationally expensive techniques like mutual information regularization can limit efficiency on high-dimensional or large-scale datasets.

Conditional diffusion models extend this idea by incorporating extra conditional information to guide generation. Meanwhile, robust learning for conditional diffusion is less explored. Na et al. (2024) represent the conditional score of noisy conditions as a linear combination of clean condition scores, weighted by instance-specific and time-dependent condition transition matrices. By minimizing the distance between the weighted score network output and the noisy data score, the diffusion model is guided to produce outputs closer to clean conditions.

Another recent study (Dufour et al., 2024) introduces a coherence score to represent the consistency between condition and demonstration, incorporating it into the diffusion model to dynamically adjust reliance on conditions. It is important to note that this method is based on a different setting: it assumes the availability of additional robust classifiers, while our work addresses robust conditional diffusion pre-training without requiring any auxiliary condition information (clean conditions in the entire training phase).

## 3 NOTATION

We summarize the main symbols used throughout the paper in Table 4 for better readability.

## 4 DERIVATION OF FORMULAS

### 4.1 RDC SDEs

Based on the standard forward and reverse-time SDEs in diffusion introduced in Section 2.1 (Eq. 1 and Eq. 2), we construct a reverse-time diffusion process for the pseudo condition $\hat{y}$ that shares the same schedule but in opposite direction as $x$. We let $x$ become $\hat{y}$, set $f(\hat{y}, t) = f(x, t)$ and keep $g(t)$ unchanged, and exchange the start and end states, where the start and end states correspond to a Gaussian noise and $\hat{y}$, respectively.

From this, we can write the forward SDE of the reverse-time diffusion condition as:

$$\mathrm{d}\mathbf{w} = f(\hat{y}, t)\mathrm{d}t + g(t)\mathrm{d}\hat{y}. \tag{12}$$

By rearranging the terms, we obtain our forward SDE for RDC:

$$\text{Forward SDE: } \mathrm{d}\hat{y} = \frac{-f(\hat{y}, t)}{g(t)}\mathrm{d}t + \frac{1}{g(t)}\mathrm{d}\mathbf{w}. \tag{13}$$

Table 4: Summary of all notations.

| Symbol | Meaning |
|---|---|
| $x \in \mathbb{R}^d$ | Demonstration |
| $y \in \mathbb{R}^d$ | Clean condition |
| $\tilde{y} \in \mathbb{R}^d$ | Noisy condition |
| $\hat{y} \in \mathbb{R}^d$ | Pseudo condition (refined from $\tilde{y}$) |
| $\widetilde{D} = \{(x^{(i)}, \tilde{y}^{(i)})\}_{i=1}^n$ | Noisy training dataset |
| $p_0(x)$ | Distribution of demonstrations $x$ |
| $p_0(x\|y)$ | Clean conditional distribution |
| $\tilde{p}_0(x, \tilde{y})$ | Noisy joint distribution |
| $p_t(x_t)$ | Transition distribution of $x$ at time $t$ |
| $s_\theta(x_t, t)$ | Score network (U-Net) w.r.t. $x_t$ and $t$ |
| $s_\theta(x_t, t, y)$ | Conditional score network given condition $y$ |
| $s_{\mathrm{CFG}}(x_t, t, y)$ | Classifier-Free Guidance score |
| $s_\phi(x_t, t, \hat{y}_t)$ | Condition score network for pseudo condition |
| $q_\phi(y\|x)$ | Auxiliary network inferring pseudo condition |
| $e_\gamma(\cdot)$ | Condition encoder (for complex conditions such as images) |
| $f(x, t)$ | Drift coefficient in SDE |
| $g(t)$ | Diffusion coefficient in SDE |
| $\mathbf{w}, \bar{\mathbf{w}}$ | Standard Wiener processes |
| $T$ | Diffusion time horizon |
| $\lambda(t)$ | Weighting function for score matching loss |
| $w$ | Guidance scale in CFG |

In this forward SDE, the drift and diffusion coefficients are:

$$f'(\hat{y}, t) = \frac{-f(\hat{y}, t)}{g(t)}, \quad g'(t) = \frac{1}{g(t)}.$$

Finally, substituting these redefined coefficients into the standard reverse SDE (Eq. 2) yields the reverse SDE for RDC:

$$\text{Reverse SDE: } \mathrm{d}\hat{y} = \left( \frac{-f(\hat{y}, t)}{g(t)} - \frac{1}{g(t)^2} \nabla_{\hat{y}} \log p_t(\hat{y}) \right) \mathrm{d}t + \frac{1}{g(t)} \mathrm{d}\bar{\mathbf{w}}. \tag{14}$$

## 4.2 INTEGRAL FORM OF $\hat{y}_\phi$ (EQ. 9)

We start from the reverse-time SDE given in Eq. 8:

$$\mathrm{d}\hat{y}_t = \left( -\frac{f(\hat{y}_t, t)}{g(t)} - \frac{1}{g(t)^2} \nabla_{\hat{y}} \log p_t(\hat{y}_t) \right) \mathrm{d}t + \frac{1}{g(t)} \mathrm{d}\bar{\mathbf{w}}_t. \tag{15}$$

To obtain a deterministic probability-flow ODE we drop the Brownian term $\mathrm{d}\bar{\mathbf{w}}_t$ and replace the score by a neural network $s_\phi(\hat{y}_t, t) \approx \nabla_{\hat{y}} \log p_t(\hat{y}_t)$:

$$\frac{\mathrm{d}\hat{y}_t}{\mathrm{d}t} = -\frac{f(\hat{y}_t, t)}{g(t)} - \frac{s_\phi(\hat{y}_t, t)}{g(t)^2}. \tag{16}$$

As mentioned before, our RDC shares the same schedule but in opposite direction with the demonstration $x$, following Karras et al. (2022), both use the implementation of $f(\cdot, t) = 0$ and $g(t) = \sqrt{2t}$. The drift simplifies to

$$\frac{\mathrm{d}\hat{y}_t}{\mathrm{d}t} = -\frac{s_\phi(\hat{y}_t, t)}{2t}. \tag{17}$$

Integrating from 0 to $T$ yields the final integral form of $\hat{y}_\phi$

$$\hat{y}_\phi = \hat{y}_0 - \int_0^T \frac{s_\phi(\hat{y}_t, t)}{2t} \, \mathrm{d}t, \tag{18}$$

where $\hat{y}_0 \sim p_T(\cdot)$ is the initial noise. In practice the integral is approximated by a numerical solver such as Euler or Heun's method.

## 4.3 DERIVATION OF TIME-ANNEALED CONDITION REGULARIZATION FOR RDC

In this section, we provides the detailed derivation of the explicit Hessian regularization term, where the RDC condition noise is generated by a process inspired by the reverse-time SDE.

Given forward SDE (Eq.8) of RDC, the RDC loss $\mathcal{L}_{\mathrm{RDC}}$ is theoretically equivalent to minimizing the negative expected log-likelihood(Kingma & Gao, 2023):

$$\mathcal{L}_{\mathrm{RDC}} \approx \arg\min_\theta \left\{ \mathbb{E}_{t,x_t} \left[ -\mathbb{E}_{\hat{y}_t \sim q(\cdot|\hat{y})} \left[ \log p_\theta(x_t|\hat{y}_t) \right] \right] \right\} \tag{19}$$

To simplify notations in Eq.8, we define a new drift term $F(\hat{y}, t) = \frac{-f(\hat{y}, t)}{g(t)}$ and a new diffusion coefficient $G(t) = \frac{1}{g(t)}$. Then we consider the infinitesimal change $\Delta\hat{y} = F(\hat{y}, t)\Delta t + G(t)\Delta\mathbf{w}$ over a small time step $\Delta t$. We apply a second-order Taylor expansion to $\log p(x_t|\hat{y} + \Delta\hat{y})$ around $\hat{y}$:

$$\log p(x_t|\hat{y} + \Delta\hat{y}) \approx \log p(x_t|\hat{y}) + (\Delta\hat{y})^T \nabla_{\hat{y}} \log p(x_t|\hat{y}) + \frac{1}{2}(\Delta\hat{y})^T \mathbf{H}_{\hat{y}}(\Delta\hat{y}) \tag{20}$$

where $\mathbf{H}_{\hat{y}} = \nabla_{\hat{y}}^2 \log p(x_t|\hat{y})$. To calculate the negative expectation with respect to the SDE increment $\Delta\hat{y}$, we will consider each item separately:

**First term** This term is simply the conditional log-likelihood objective evaluated at the unperturbed condition $\hat{y}$: $-\log p(x_t|\hat{y})$.

**Second term** The expectation of this term contributes a first-order correction term:

$$-\mathbb{E}[(\Delta\hat{y})^T]\nabla_{\hat{y}} \log p \approx -(F(\hat{y})\Delta t)^T \nabla_{\hat{y}} \log p \tag{21}$$

This term is a first-order correction resulting from the SDE's drift $F(\hat{y}, t)$. We conceptually group this term with the zeroth-order term ($-\log p(x_t|\hat{y})$) because they together constitute the primary conditional diffusion objective.

**Third term** This term provides the explicit Hessian regularization. We calculate the negative expectation of the second-order Taylor term: $-\frac{1}{2}\mathbb{E}[(\Delta\hat{y})^T \mathbf{H}_{\hat{y}}(\Delta\hat{y})]$.

To compute the expectation of this quadratic form, we use the trace identity $\mathbb{E}[(\Delta\hat{y})^T \mathbf{H}(\Delta\hat{y})] = \mathrm{Tr}(\mathbf{H} \cdot \mathrm{Cov}(\Delta\hat{y}))$. The covariance of the SDE increment is determined by the diffusion coefficient: $\mathrm{Cov}(\Delta\hat{y}) = G(t)^2 \mathbf{I}\Delta t = \frac{1}{g(t)^2}\mathbf{I}\Delta t$. Applying the trace identity yields:

$$-\frac{1}{2}\mathbb{E}[(\Delta\hat{y})^T \mathbf{H}_{\hat{y}}(\Delta\hat{y})] \approx -\frac{1}{2}\mathrm{Tr}(\mathbf{H}_{\hat{y}} \cdot \mathrm{Cov}(\Delta\hat{y}))$$

$$= -\frac{1}{2}\mathrm{Tr}\left(\mathbf{H}_{\hat{y}} \cdot \frac{1}{g(t)^2}\mathbf{I}\Delta t\right)$$

$$= -\frac{1}{2g(t)^2}\Delta t \cdot \mathrm{Tr}(\mathbf{H}_{\hat{y}})$$

We can finally define the full RDC objective as:

$$\mathcal{L}_{\mathrm{RDC}} \approx \arg\min_\theta \left\{ -\mathbb{E}_{t,x_t} \left[ \log p_\theta(x_t|\hat{y}) \right] + \mathbb{E}_{t,x_t} \left[ \frac{1}{2g(t)^2}\Delta t \cdot \mathrm{Tr}(\nabla_{\hat{y}}^2 \log p_\theta(x_t|\hat{y})) \right] \right\}, \tag{22}$$

where the second term is a time-annealed regularization penalizing the sensitivity of the conditional log-likelihood to condition noise(sum of second-order derivatives over all condition dimensions).

## 5 MORE DETAILS OF EXPERIMENTS

### 5.1 NOISY CONDITION SETTING

1) **Visuomotor Policy Generation**

We conduct experiments on this task with Push-T dataset. Push-T is an object pushing task for robots: The robotic arm needs to push a gray T-shaped block at a random position to the green target position based on the image observations. This dataset contains 206 video clips (25650 frames).

In this task, the image observation is the condition, radial distortion and tangential distortion are two typical forms of lens distortion for image data, which are widely exist in the actual imaging process. We simulate these two types of image noise with the following two expressions. Set $(x, y)$ as the coordinates of the image.

- Radial Distortion. Radial distortion is mainly caused by the non ideal characteristics of the lens optical components, resulting in varying degrees of distortion of the image edges relative to the center, presenting as barrel or pillow distortion.

$$\begin{cases} x' = x(1 + k_1 r^2 + k_2 r^4), \\ y' = y(1 + k_1 r^2 + k_2 r^4), \end{cases} \tag{23}$$

  where $k_1$ and $k_2$ are the first- and second-order radial distortion coefficients that control the distortion intensity. In close-up tasks such as robot grasping of objects, we set $k_1 \in [0, \ 0.2]$ and $k_2 = 0$ to simulate barrel distortion during the close-up process.

- Tangential Distortion. Tangential distortion is caused by improper installation between the lens and imaging sensor, manifested as a deviation of the image in a certain direction.

$$\begin{cases} x' = x + 2p_1 xy + p_2(r^2 + 2x^2), \\ y' = y + 2p_1(r^2 + 2y^2) + p_2 xy, \end{cases} \tag{24}$$

  where $p_1$ controls the distortion along the x-direction, and $p_1$ controls distortion along the y-direction. We set $p_1 \in [0, \ 0.1]$ and $p_2 \in [0, \ 0.1]$.

  We further introduce spatial noise to simulate minor geometric disturbances of the camera sensor position. Specifically, we perturb the coordinates by adding independent pixel-wise offsets to the horizontal and vertical directions:

$$\begin{cases} x' = x + \Delta x, \\ y' = y + \Delta y, \end{cases} \tag{25}$$

  where $\Delta x$ and $\Delta y$ represent the added noise in the x and y directions, respectively. These values are randomly sampled within the ranges $\Delta x \in [0, \ 10]$ and $\Delta y \in [0, \ 10]$.

As shown in Figure 5, we visualize several examples from the Push-T dataset before and after the application of the above noise. Each pair of images illustrates how camera distortion and spatial perturbations affect the image observation of the same scene. These visualizations help demonstrate challenges in robotic perception under the close-range operation.

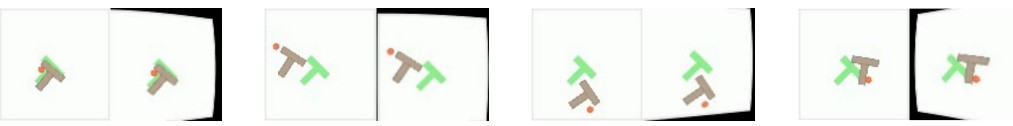

Figure 5: Visualization examples of noisy Push-T dataset. Each subfigure shows the effect of condition noise, where the left image is the original image observations and the right image is the corresponding version with added camera distortion.

2) **Image Generation**

We conduct image generation experiments on both CIFAR-10 and CIFAR-100 datasets. CIFAR-10 consists of 60000 32x32 pixel color images, divided into 10 classes, with each class containing 6000 images. These images cover classes such as airplanes, cars, birds, cats, deer, dogs, frogs, horses, boats, and trucks. CIFAR-10 is suitable for label-condition image generation task. CIFAR-100 is similar to CIFAR-10, but contains 100 classes, each class containing 600 images, for a total of 60000 images. These classes include various fine-grained objects, such as different types of fish, insects, flowers, etc. In our setting, CIFAR-100 is used to evaluate the robust condition generation performance of the model when dealing with more complex categories.

In the image generation task, the class-label is the condition. In order to better cope with label uncertainty in real-world scenarios, we introduced symmetric noise and asymmetric noise into both the CIFAR-10 and CIFAR-100 dataset.

- Symmetric Noise: symmetric noise refers to the situation where the label of each class is mislabeled as another class with the same probability $\eta$, simulating a random, unbiased label error scenario.
- Asymmetric Noise: asymmetric noise refers to the tendency of certain classes to be mistakenly labeled as other specific classes based on their similarity and other characteristics. For example, images of "cats" are mistakenly labeled as "dogs" instead of random other classes. This type of noise can better reflect label confusion caused by visual similarity and other factors in actual scenes. For CIFAR-10, labels are flipped by truck $\leftrightarrow$ automobile, bird $\leftrightarrow$ airplane, deer $\leftrightarrow$ horse, cat $\leftrightarrow$ dog. For CIFAR-100, classes are randomly flipped within the same superclass.

## 5.2 DETAILS OF PREDICTION HEAD

In the label-conditioned image generation task, the prediction head consists of two convolutional layers with batch normalization and a residual connection, followed by a global average pooling layer and a final fully connected linear classifier.

In the image-condition visuomotor policy generation task, the prediction head processes the 1D image embedding via a 1D convolution followed by temporal pooling and flattening.

## 5.3 DETAILS OF EXPERIMENTAL SETUP

**2-D Data Generation** For the 2-D toy case, a 1-D U-Net is adopted as the score matching network. The model is trained with a batch size of 512 for a total of 10,000 iterations. For the early stopping, we set $0 \sim 500$ iterations. All other hyperparameters are kept consistent across different noise levels. For the sampling process, we use deterministic sampling with Heun 2nd-order integrator; noise schedule $\sigma(t) = t$, signal scaling $s(t) = 1$; step-density exponent $\rho = 7$; 35 Number of Function Evaluations (NFE); $\sigma_{\min} = 0.002$, $\sigma_{\max} = 80$ (Karras et al., 2022).

**Conditional Image Generation** In this section, we utilized 8 NVIDIA Tesla 4090 GPUs and employed CUDA 11.8 and PyTorch 2.0 versions in our experiments. Our model framework and code are based on EDM (Karras et al., 2022). For all experiments, we used DDPM++ network architecture with a U-net backbone, which is originally proposed by Song et al. (2021b) and modified by Karras et al. (2022). The training setting is the same with Karras et al. (2022). Besides that, we set the $\alpha$ in the Eq. 7 as 0.1. For the early stopping, we set $0 \sim 25,000$ iterations for CIFAR-10 and $0 \sim 30,000$ iterations for CIFAR-100 under the symmetric noise setting.

For the sampling process, we use deterministic sampling with Heun 2nd-order integrator; noise schedule $\sigma(t) = t$, signal scaling $s(t) = 1$; step-density exponent $\rho = 7$; 35 NFE; $\sigma_{\min} = 0.002$, $\sigma_{\max} = 80$, which is exactly the setting of Karras et al. (2022). Besides that, we apply the following metrics to evaluate our method:

- **FID** ($\downarrow$) evaluates the quality of generated images by computing the Fréchet distance between the feature distributions of generated and given reference images. Smaller FID indicates higher generation quality.
- **IS** ($\uparrow$) measures the quality and diversity of the generated image by using a pretrained Inception model to predict the generated image, and calculating the KL divergence of the predicted distribution of the generated image.

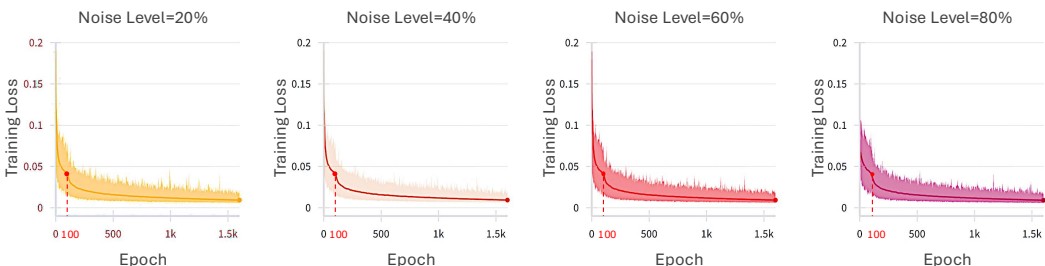

Figure 6: Visuomotor policy training loss over epochs under 20%–80% camera distortion. The loss drops sharply before entering a plateau, corresponding to the transition from fitting the clean observation to memorizing noisy one. Early stopping is applied at the beginning of this plateau.

- **Density** (↑) reflects the distribution density of generated images in the feature space, revealing the concentration and coverage of generation. High density value reveals a concentrated distribution in the feature space, with better consistency and representativeness.

- **Coverage** (↑) measures the extent a generative model covers the distribution of the given reference distribution. High coverage value indicates that the model can comprehensively cover the distribution of the given reference distribution and generated images are more diverse and representative.

- **CW (Class-wise) Metrics** compute the average value of the metric within each class. For example, **CW-FID** (↓) reflects the generation quality of the model on each class, and **CW-Density** (↑) and **CW-Coverage** (↑) evaluate the distribution density and coverage within each class, respectively.

**Visuomotor Policy Generation** In this section, all experiments are conducted on 3*A100 GPUs using the AdamW optimizer. The learning rate is set to 1e-4, with a weight decay rate of 1e-6. The model is trained for 3000 epochs with a batch size of 64, and we repeat our experiments on the random seeds of 42, 43, and 44. Target Area Coverage (TAC) metric is used to measure the quality of visuomotor policy, which is to compute the IoU between the T-shaped block and the green target area after all the operation of the robotic arm. Besides that, we set the $\alpha$ in the Eq. 7 as 0.2, and the early stopping is set as $0 \sim 100$ epochs.

### 5.4 ANALYSIS OF EARLY STOPPING FOR PSEUDO-CONDITION LEARNING

Early stopping is applied to prevent pseudo-conditions from overfitting to noisy observations. Due to the memorization effect, models first fit clean condition samples (easy) and later overfit noisy ones (hard), producing a loss curve that drops sharply before plateauing (Figure **??**). We freeze pseudo-condition updates once the loss stabilizes; temporal ensembling further smooths the updates, making the diffusion model largely insensitive to the exact stopping point.

Table 5 reports visuomotor policy generation performance (TAC) under 80% camera distortion with different early stopping epochs. The results are stable across a range of epochs, demonstrating the robustness of this procedure.

Table 5: Visuomotor policy generation performance of our method with different early stopping epochs under 80% camera distortion.

| Early Stopping Epoch | TAC |
|:---:|:---:|
| 80 | $72.55 \pm 1.78$ |
| 100 | $71.78 \pm 3.24$ |
| 120 | $72.65 \pm 1.49$ |

## 5.5 Image Generation Conditioned on Labels with Asymmetric Noise

In this section, we tested the label-condition image generation performance of our method under a more challenging setting, which is to simulate condition confusion between similar classes. Specifically, we introduced asymmetric noise of 20% and 40% on both the CIFAR-10 and CIFAR-100 datasets. Here, we define the similar classes the same with Na et al. (2024), which is clearly stated in Appendix 5.1

For all the experiments under asymmetric noise, we follow the training and sampling settings of Karras et al. (2022) as well. Besides that, we set the $\alpha$ in the Eq. 7 as 0.3 for CIFAR-10 and 0.5 for CIFAR-10. For the early stopping, we set $0 \sim 25,000$ iterations for CIFAR-10 and CIFAR-100.

As shown in the Table 6, even though under the case of 40% asymmetric noise on CIFAR-100, our method demonstrates comparable performance to the baseline method, the performance of our method far exceeds that of SOTA method TDSM in all other settings.

Table 6: Conditional Generation Performance Comparison with EDM and TDSM on the CIFAR-10 and CIFAR-100 datasets under Asymmetric Noise.

| Dataset | Noise Level | Method | FID ($\downarrow$) | IS ($\uparrow$) | Density ($\uparrow$) | Coverage ($\uparrow$) | CW-FID ($\downarrow$) | CW-Density ($\uparrow$) | CW-Coverage ($\uparrow$) |
|---|---|---|---|---|---|---|---|---|---|
| **CIFAR-10** | 0% | EDM | 1.92 | 10.03 | 103.08 | 81.90 | 10.23 | 102.63 | 81.57 |
| | 20% | EDM | **2.02** | 10.06 | 100.66 | 81.36 | 11.97 | 96.10 | 79.95 |
| | | TDSM | **1.95** | 10.04 | 104.15 | 81.81 | 10.89 | 101.77 | 80.99 |
| | | Ours | 2.17 | **9.97** | **105.59** | **93.72** | **6.8** | **103.31** | **93.32** |
| | 40% | EDM | **2.23** | 10.09 | 101.25 | 81.10 | 15.18 | 92.13 | 78.12 |
| | | TDSM | **2.06** | 10.02 | **105.19** | 81.90 | 12.54 | 99.21 | 79.98 |
| | | **Ours** | 2.14 | **10.02** | 103.15 | **93.02** | **12.47** | **99.25** | **91.79** |
| **CIFAR-100** | 0% | EDM | 2.51 | 12.80 | 87.98 | 77.63 | 66.97 | 82.58 | 75.78 |
| | 20% | EDM | **2.76** | 12.49 | 87.36 | 77.04 | 75.39 | 33.31 | 72.14 |
| | | TDSM | 2.64 | 12.79 | 88.41 | 77.46 | **69.83** | 78.92 | 74.01 |
| | | **Ours** | 3.34 | **12.92** | **98.43** | **91.98** | 66.89 | **84.95** | **90.57** |
| | 40% | EDM | **2.73** | 12.51 | **87.06** | **76.56** | 89.13 | 60.27 | 64.19 |
| | | TDSM | 2.81 | **12.57** | 87.01 | 76.27 | **73.13** | **74.30** | **71.48** |
| | | Ours | 2.83 | 12.51 | 86.74 | 75.88 | 89.34 | 59.69 | 63.33 |

## 5.6 Image Generation Conditioned on Semantic Maps

We evaluate our method on semantic synthesis (Rombach et al., 2022) to explore its potential for augmenting medical image semantic datasets. In medical imaging, precise mask annotations are crucial, as boundary noise from ambiguous pathological regions can mislead model training and even contribute to misdiagnosis.

To simulate such noisy condition, we perturb the mask annotations of the ISIC 2018 (Codella et al., 2019) dataset with a two-pixel erosion or dilation, introducing varying levels of boundary noise. Our generation model is trained on these noisy masks and later tested with clean masks as input. For comparison, we additionally use DuAT (Tang et al., 2023), a advanced medical image mask prediction method with 86% mIoU accuracy on the ISIC 2018 (Codella et al., 2019) dataset, to provide realistic predicted masks for evaluation.

We assess both the photorealism and the mask consistency of the generated results using mean IoU (mIoU) and mean Dice coefficient (mDice). Specifically, mIoU measures the average overlap between predicted and clean mask regions across classes, while mDice captures the harmonic similarity with greater sensitivity to boundary alignment.

Table 7 summarizes the quantitative results, showing that our current implementation underperforms the baseline methods. This indicates that the model in its current form is not yet capable of effectively handling boundary noise. To better understand this behavior, we further plot the Mask Fitting Speed Comparison during Training (Figure 7). The figure reveals that, in the early stages of training, our method exhibits a desirable effect: the pseudo-condition aligns more closely with the clean mask than with the noisy mask. However, the model quickly overfits to the noisy annotations, and this rapid convergence prevents us from leveraging the early-stage advantage for effective correction.

Table 7: Mean IoU Comparison with EDM on the ISIC 2018 datasets under Boundary Noise.

| Noise Level | Method | mIoU | mDice |
|:---:|:---:|:---:|:---:|
| 0% | EDM | **86.56** | **92.67** |
| 20% | EDM | **84.82** | **91.62** |
|  | **Ours** | 84.20 | 91.24 |
| 40% | EDM | **83.86** | **91.03** |
|  | **Ours** | 83.21 | 90.63 |
| 60% | EDM | 82.14 | 89.96 |
|  | **Ours** | **82.94** | **90.47** |
| 80% | EDM | **80.73** | **89.07** |
|  | **Ours** | 80.11 | 88.68 |

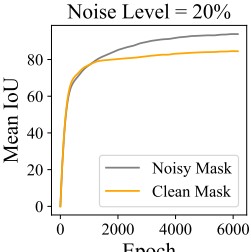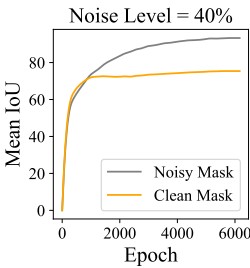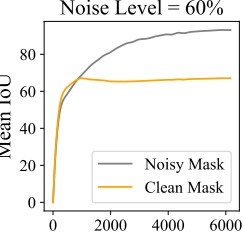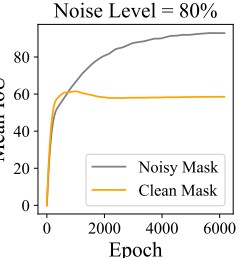

Figure 7: Mask Fitting Speed Comparison during Training. Each subplot shows the mean IoU between our predicted Pseudo Condition and the observed noisy mask (orange line) or the unobserved clean mask (gray line) over training epochs. From left to right, the subplots correspond to different noise levels from 20% to 80%.

## 5.7 VISUOMOTOR POLICY GENERATION CONDITIONED ON IMAGE WITH GAUSSIAN NOISE

In this section, we provide additional experiments to evaluate the robustness of our method under more realistic noise conditions. Specifically, we inject extreme (60% and 80%) Gaussian noise into the image observations. As shown in Table 8, our method maintains strong performance under both noise levels, highlighting its robustness when handling gaussian noise images.

Table 8: TAC Comparison with DP on the Push-T dataset under gaussian noise.

| Method | Noise Level | |
|:---:|:---:|:---:|
|  | 60% | 80% |
| DP | 80.32±2.66 | 79.58±1.51 |
| **Ours** | **81.47±1.36** | **81.76±0.62** |

## 5.8 IMAGE GENERATION CONDITIONED ON LABEL UNDER TRANSFORMER BACKBONE

In this section, we further examine the generality of our method by replacing the UNet backbone with a Transformer-based architecture. For the Transformer backbone, we strictly followed the implementation of DP (Vaswani et al., 2017). The design of the condition prediction head remains unchanged from the implementation described in Appendix 5.2. This experiment evaluate whether our method remains effective under a different diffusion model backbone. As shown in Table 9, our method consistently outperforms DP when conditioned on 60% and 80% image observation distortion, demonstrating that the improvements of our method are not tied to a specific network architecture.

Table 9: TAC Comparison with DP on the Push-T dataset under Transformer Backbone.

| Method | Noise Level | |
| --- | --- | --- |
| | 60% | 80% |
| DP | 60.45±2.70 | 60.54±**1.94** |
| **Ours** | **62.72±0.87** | **62.19**±3.47 |

## 5.9 VISUALIZATION RESULTS ON LABEL-CONDITION IMAGE GENERATION

To demonstrate the superiority of our method more intuitively, under the CIFAR-10 training setting of $40\%$ symmetric noise, we randomly sampled a noise and generated it under exactly the same conditions using the current SOTA method TDSM (Na et al., 2024) and our method, and visually compared the generation effects of the two methods. As shown in the Figure 8, we randomly present two example results. It can be seen that under $40\%$ symmetric noise, conditional image generation using TDSM (Na et al., 2024) sometimes fails to generate certain classes of images and produces some unrecognizable content, while our method can robustly generate images that meet the requirements of each class.

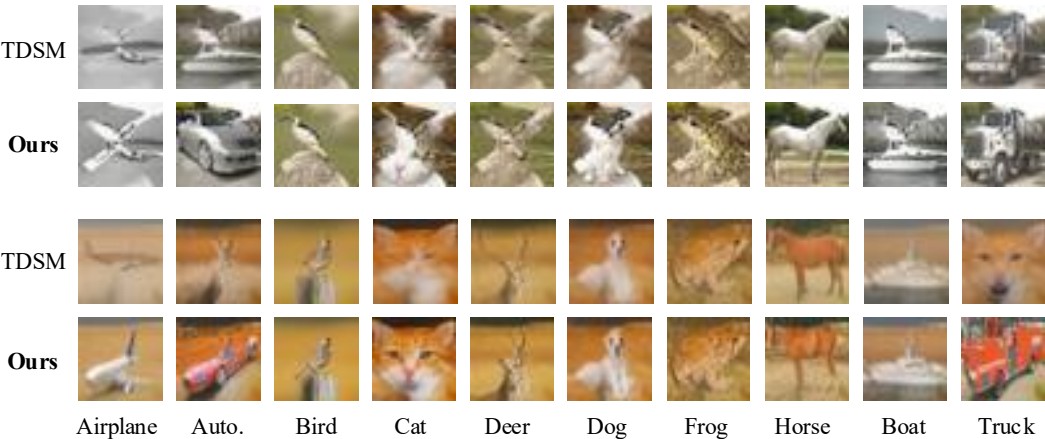

Figure 8: Visualization examples on CIFAR-10 compared with TDSM under $40\%$ symmetric noise, where "auto." is short for "automobile".

Besides the above results, we provide more visualization results of our method on CIFAR-100 under $60\%$ symmetric noise. As shown in Figure 9, we trained our model under $60\%$ symmetric noise training set, and then sampled seven different typical classes (apple, goldfish, bear, bed, beetle, bicycle, and bottle) starting from the same 32 random noise images. This figure shows the label-conditioned image generation performance of our method under different conditions given the same initial state. Figure 9 shows that our method can maintain robust conditional generation performance even under extreme $60\%$ symmetric noise.

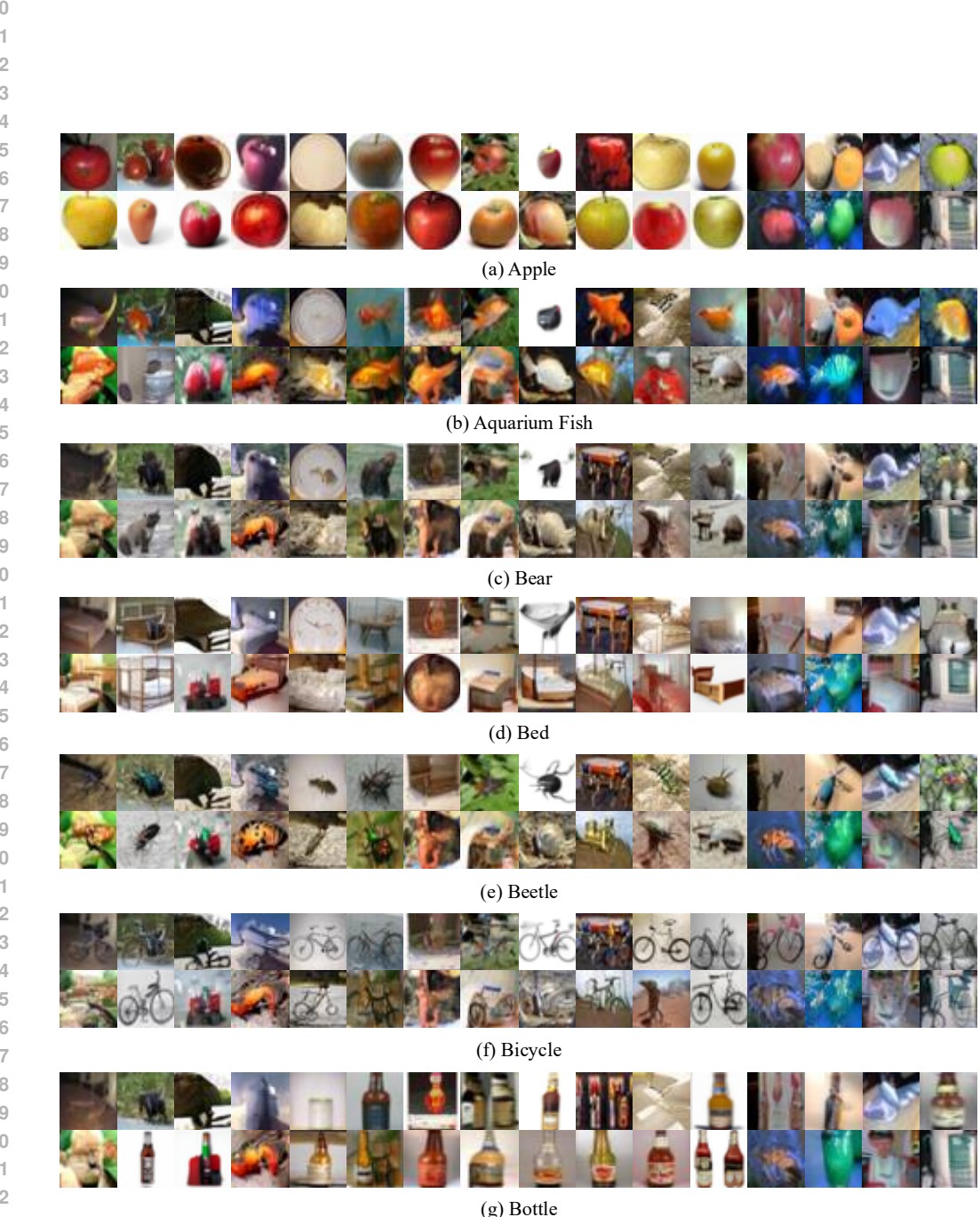

(a) Apple

(b) Aquarium Fish

(c) Bear

(d) Bed

(e) Beetle

(f) Bicycle

(g) Bottle

Figure 9: Visualization examples of our method on CIFAR-100 under 60% symmetric noise.

