# OpenReview forum: "Robust Learning of Diffusion Models with Extremely Noisy Conditions"
_ICLR.cc/2026/Conference — Submitted to ICLR 2026_

### Official Review · Reviewer_FovT · 2025-10-30

**Soundness:** 3
**Presentation:** 3
**Contribution:** 3
**Rating:** 6
**Confidence:** 3

**Summary:**

This paper introduces a robust learning framework for conditional diffusion models under extremely noisy conditions, such as corrupted labels or unreliable observations. The authors identify that existing noise-robust methods fail when noise levels are high, and propose two key innovations:

Pseudo Conditions (PC): A surrogate for clean conditions, initialized from noisy ones and progressively refined using temporal ensembling and early stopping to avoid overfitting.

Reverse-time Diffusion Condition (RDC): A technique that applies the diffusion process to pseudo conditions in reverse, enhancing memorization and stabilizing training under noise.

The method is evaluated on class-conditional image generation (CIFAR-10/100) and visuomotor policy generation (Push-T), achieving state-of-the-art performance across various noise levels. Ablation studies confirm the effectiveness of both PC and RDC components.

**Strengths:**

The paper introduces a genuinely new role for diffusion dynamics: instead of diffusing only the image, it diffuses the condition itself in reverse time (RDC). This is not a minor tweak; it reframes the noisy-label problem as a joint denoising task in both pixel and label space and gives the diffusion model a self-contained way to “hallucinate then refine” its own conditioning signal.

The combination of (i) learning a pseudo-condition, (ii) updating it via temporal ensembling, and (iii) injecting it back through an RDC augmentation has not appeared before in the diffusion literature. Even individually, these ideas are creatively re-purposed from semi-supervised learning, mean-teacher methods, and score-based generative models.

The work removes a key limitation of prior label-noise diffusion papers (TDSM, RCGAN, etc.)—the need to estimate an explicit noise-transition matrix—and still works when 80 % of the labels are adversarially corrupted.

The method is model-agnostic: any conditional diffusion backbone (U-Net, DiT, etc.) can slot in the lightweight prediction head and RDC loss. This makes the barrier to adoption low for practitioners who already have diffusion pipelines.

 It opens a new research direction—self-correcting conditional diffusion—that could extend to text-to-image (noisy captions), reinforcement learning (noisy rewards), or audio (transcript errors).

**Weaknesses:**

The paper motivates RDC by analogy to “data augmentation with Gaussian noise” and by an intuitive SDE reversal, but it never proves that the reverse-time diffusion of the condition actually improves the denoising error or the posterior p(clean-label | noisy-label).


Training freezes the pseudo-label at 25 k (CIFAR-10) or 30 k (CIFAR-100) iterations—numbers taken from a grid search on clean validation data. In real noisy-data scenarios we do not have clean validation labels.

**Questions:**

Refer to Weaknesses

---

> ### Author Response · Authors · 2025-11-27
> **Official Comment by Author**
>
> Thank you for your comments!
>
> ## Weaknesses:
> > The paper motivates RDC by analogy to “data augmentation with Gaussian noise” and by an intuitive SDE reversal, but it never proves that the reverse-time diffusion of the condition actually improves the denoising error or the posterior p(clean-label | noisy-label).
>
>
> RDC is designed for pseudo-condition augmentation in early-stage training, rather than for building a dual “condition + demonstration’’ diffusion model. To make this clear, we analyze in Sec. 3.1.2 and Appendix 4.3 that RDC is mathematically equivalent to adding a time-annealed regularizer that penalizes the sensitivity of the conditional log-likelihood to condition noise, thereby enhancing the learning of pseudo-conditions.
>
>
> > Training freezes the pseudo-label at 25 k (CIFAR-10) or 30 k (CIFAR-100) iterations—numbers taken from a grid search on clean validation data. In real noisy-data scenarios we do not have clean validation labels.
> RDC is designed simply to strengthen the pseudo-conditional signal; the diffusion refinement is only applied before early stopping, after which training proceeds normally. Thus we do not aim to guarantee improvements in the exact reverse-time denoising error, but rely on the empirical observation—consistent with [1]—that diffusion perturbation enhances supervision under noisy labels.
>
> Pseudo-condition updates are stopped based on loss stabilization, not clean validation data, ensuring a stable and reproducible procedure. Due to the memorization effect, models fit easy (clean) samples first and later overfit hard (noisy) ones, producing a loss curve that drops sharply then plateaus. Temporal ensembling smooths pseudo-condition updates, making the diffusion model largely insensitive to the exact stopping point.

---

### Official Review · Reviewer_oTa3 · 2025-10-30

**Soundness:** 2
**Presentation:** 2
**Contribution:** 2
**Rating:** 2
**Confidence:** 4

**Summary:**

## Review of “ROBUST LEARNING OF DIFFUSION MODELS WITH EXTREMELY NOISY CONDITIONS”

### Summary
The paper studies conditional diffusion models when the conditioning signal (labels for class-conditional image generation, visual observations for visuomotor policy generation) is extremely noisy. The authors argue existing robustness methods fail at high noise rates and propose two ideas:
1. **Pseudo condition (PC):** learn a per-sample latent “pseudo condition” \(\hat y\) using a lightweight prediction head and temporal ensembling, then gradually replace the noisy condition \(\tilde y\) with \(\hat y\).
2. **Reverse-time Diffusion Condition (RDC):** inject noise into \(\hat y\) via a reverse-time SDE so that the model must learn to generate conditioned on progressively denoised pseudo conditions.

They claim state-of-the-art performance on CIFAR-10/100 with up to 80% label noise and on a visuomotor policy learning benchmark with distorted camera inputs.

**Strengths:**

The paper targets a setting that is undeniably important: conditional diffusion models trained with highly corrupted conditioning signals, including 60–80% incorrect class labels on CIFAR-style datasets and visual inputs with heavy distortion in visuomotor tasks. Even if the proposed solutions need more clarity, the problem itself is real and underexplored at the level of “extremely noisy conditions,” rather than just mild label noise. The paper does a good job motivating why this matters for reliability and safety.

**Weaknesses:**

### Major Concerns

#### 1. Conceptual clarity / internal consistency
The core story of the paper is: (i) noisy conditions \(\tilde y\) entangle clusters, (ii) we “repair” them by learning pseudo conditions \(\hat y\), (iii) we further stabilize training using RDC. But the paper never really gives a coherent probabilistic view of what \(\hat y\) is supposed to represent.

- This is just classic self-training / bootstrapping on noisy labels with EMA. There is no guarantee provided about to where the \(\hat y\) converges rather than to whichever early bias the model latches onto.

- The authors appeal to “memorization effect” and “early stopping” but give no principled stopping rule, no theoretical identifiability, and no robustness analysis under heavy class imbalance or systematic (non-symmetric) corruption. This is hand-wavy.

- Later, \(\hat y\) becomes part of an SDE and is diffused in reverse time (Sec. 3.1.2). But now \(\hat y_t\) is treated almost like a continuous signal you can inject noise into, including for discrete labels. The paper never reconciles these two roles: is \(\hat y\) a discrete class label, a continuous embedding, or a generic latent code? The text casually mixes these cases, e.g. in label-conditioning vs. visuomotor conditioning, and the math assumes a continuous vector with gradients \(\nabla_{\hat y} \log p_t(\hat y)\). This looks mathematically incompatible with the categorical-label case that motivates most of the “extremely noisy label” narrative.

- The RDC SDE in Eq. (8) is introduced with forward / reverse SDEs copied from diffusion literature, but the derivation is not convincing. The paper defines boundary conditions \( \hat y_0 \sim \mathcal{N}(\mu,\sigma) \), \( \hat y_T = \hat y \), then writes down dynamics that allegedly realize a “reverse-time diffusion condition.” However:
  - There is no demonstration that these dynamics actually produce a tractable marginal \(p_t(\hat y)\) consistent with those boundary conditions.
  - There is no training algorithm that simulates these SDEs in practice in a numerically well-defined way beyond saying “we estimate \(\hat y_\phi\) using numerical methods.” This is extremely vague.

In short: the paper sells RDC as a principled diffusion-in-condition-space, but the derivation looks more like informal noise injection / augmentation of the conditioning vector. The gap between the math and the actual implementation is huge and currently not bridged.

#### 2. Method description is incomplete / borderline non-reproducible
Several critical training details are missing or contradictory:

- **How exactly is \(\hat y\) updated?**
  The algorithmic description for pseudo condition learning (Eq. (6)–(7), Algorithm 1) is ambiguous:
  - Eq. (6) optimizes \(\| \hat y_\phi - \tilde y \|^2\). But if \(\tilde y\) is extremely noisy, regressing toward it just propagates noise. Why should this denoise anything, instead of *fitting the noise faster*? The only answer given is “memorization effect” / “temporal ensembling” / “early stopping,” but the paper does not specify (a) the EMA momentum \(\alpha\), (b) when to stop, or (c) how sensitive results are to those choices.
  - Algorithm 1 line 7 says `yt ← (1−λ)yt + λ ŷϕ`. This quietly introduces a new \(\lambda\) and mutates \(y_t\) in-place, but \(\lambda\) is never defined, nor is it connected to \(\alpha\) in Eq. (7). This raises reproducibility and even correctness concerns.

- **When is the condition encoder trained and frozen?**
  For image-based conditions (visuomotor setting), the paper says: initialize \(\hat y\) with encoder output \(\tilde y = e_\gamma(\text{image})\); run early stopping to refine \(\hat y\); later continue training the encoder to match \(\hat y\) (Sec. 3.2).
  This implies a bilevel schedule (first fix \(e_\gamma\), tune \(\hat y\); then fix \(\hat y\), tune \(e_\gamma\)). But:
  - There is no precise schedule (epochs? steps? loss plateaus? validation metric?).
  - There is no ablation isolating whether this two-phase procedure, *not* RDC, is what actually gives the reported visuomotor gains.

- **Sampling / guidance**
  The method claims to “switch” classifier-free guidance from \(\tilde y\) to \(\hat y\) in Eq. (5). But at sampling time you don’t have ground-truth clean labels or clean observations — you either have the noisy condition \(\tilde y\) (test-time corruption) or nothing. The paper doesn’t say how \(\hat y\) is obtained at inference. Do we run the prediction head \(q_\phi\) online to predict a denoised condition at test time? Do we rely on a refined encoder \(e_\gamma\)? This is crucial for deployment, especially in the robotics/control setting, and it’s missing.

Overall, too many essential knobs are “empirically determined,” “estimated numerically,” or “updated via early stopping,” without concrete, reproducible definitions. For ICLR-level work, that’s not acceptable.

#### 3. Weak/unclear baselines and metrics
The empirical claims are not convincingly supported.

- **Label-noise baselines.**
  The paper mostly compares against EDM (Karras et al., 2022) and TDSM (Na et al., 2024). But robustness to label noise has an extensive literature: MentorNet, Co-teaching, DivideMix, early-learning regularization, transition-matrix estimation, etc. Many of those works explicitly address extreme label noise and could be adapted to conditional generative models or to the conditional encoder. The paper cites some of them in Related Work but does not actually implement competitive versions (e.g. using DivideMix-style clean/noisy split on the condition head, or robust loss on the condition head rather than plain \(\ell_2\)). So “SOTA” is overstated.

- **Metric choice for controllability.**
  In Figure 1(b), controllability is measured as top-1 accuracy of a pretrained CIFAR-10 classifier on generated samples.
  This is fragile: if the classifier itself struggles under heavy corruption or distribution shift, the “controllability” score will be noisy. No calibration is provided. Also, top-1 accuracy on generated samples says nothing about sample diversity or mode-collapse. You *could* cheat controllability by collapsing to a single prototypical “dog” image per class. The paper does include FID/IS/Density/Coverage in Table 1, but does not analyze mode collapse explicitly in the high-noise regime (60–80%), which is exactly where they claim superiority.

- **CIFAR-100 @ 60–80% noise.**
  The paper shows very large claimed gains when noise is extremely high and TDSM “collapses,” and then takes that as evidence of novelty. But TDSM is originally designed for label noise via transition matrices and might simply be mis-implemented for 100-way classification under 80% corruption, which is an extremely adversarial setting. There is no sanity check like: what is the oracle upper bound if you just train on clean labels for a small clean subset? What if you partially relabel with a small trusted clean set? Without that, “SOTA” here mostly means “our baseline impl of TDSM crashes so we win.”

- **Visuomotor / robotics experiments.**
  The robotics evaluation is (i) one environment (Push-T), (ii) one type of corruption (camera distortion with fixed probability), and (iii) one metric (IoU/TAC).
  This is extremely narrow. There’s no evaluation of closed-loop robustness under *unseen* distortions, lighting changes, occlusions, or partial sensor dropout — which is exactly the kind of real-world brittleness the introduction uses to motivate safety (“autonomous driving,” “surgery,” etc.). The leap from “slightly distorted tabletop pushing in simulation” to “hazardous failures in autonomous driving and surgery” is not justified.

In short: the experiments are tailored to showcase the authors’ method, but they do not seriously test robustness, and they omit strong alternative baselines.

#### 4. Mathematical rigor of RDC
RDC is pitched as a key novelty, but right now it reads like ad hoc noise augmentation with diffusion-flavored notation:

- Eq. (8) writes forward and reverse SDEs for \(\hat y\), but there is no derivation that these correspond to an ELBO-style bound, or that optimizing Eq. (10) is consistent with score matching in \(\hat y\)-space. The connection to Kingma & Gao (2023) is waved at but not actually developed.

- Eq. (9) defines \(\hat y_\phi\) as an integral of the learned score function over \(t\). But then the paper immediately says “we directly optimize \(\|\hat y_\phi - \tilde y\|^2\)” instead of training the score network in \(\hat y\)-space in a principled way. This looks self-contradictory: either RDC gives you a principled score-matching view, or you’re just using it as data augmentation for a regression head.

- The ablation in Table 3 is used to argue RDC “fixes” the degradation caused by naïve pseudo conditions. But that table is only on CIFAR-10 40% noise, and it is extremely underspecified: hyperparameters, stopping rules, and sampling details are not given. So it’s impossible to tell whether RDC itself is doing anything fundamental, or whether the improvement is due to other training heuristics that were added alongside it (e.g., different EMA schedule or guidance scaling).

Given how much of the claimed novelty rests on RDC, this level of vagueness is a serious issue.

**Questions:**

Do we really need the REVERSE-TIME DIFFUSION CONDITION? is that possible to define a manual schedule to anneal the noise level of the \hat y. I think here the author lacks enough discussion about the intuition/motivation of the introducing of such a techinique.

---

> ### Author Response · Authors · 2025-11-27
> **Official Comment by Author**
>
> Thank you for your comments!
>
> ## Weaknesses:
> ### Major Concerns
> **1. Conceptual clarity / internal consistency**
> > The core story of the paper is: (i) noisy conditions (\tilde y) entangle clusters, (ii) we “repair” them by learning pseudo conditions (\hat y), (iii) we further stabilize training using RDC. …
>
> Self-training/bootstrapping with EMA handles $f(x) \rightarrow y$, whereas our method deals with $f(x,y) \rightarrow {\nabla_{y} \log p_t(y)}$, reflecting a fundamental structural difference between the two approaches. The pseudo-conditions we predict are used as conditional inputs to update the diffuison model rather than as supervision signals, which explains why our method does not suffer from early-bias.
>
>
> > The authors appeal to “memorization effect” and “early stopping” but give no principled stopping rule, no theoretical identifiability…
>
> As discussed in Appendix 5.4, we stop pseudo-condition updates once the training loss stabilizes. While we currently rely on empirical early stopping, developing a principled stopping rule is left for future work.
> Our findings are primarily empirical, based on the memorization effect rather than theoretical identifiability. Class imbalance is outside the scope of this work. We did evaluate systematic (non-symmetric) corruption, as detailed on P23 and Appendix 5.4, showing robustness under 20% and 40% asymmetric noise. Note that 40% asymmetric noise is already extreme—if class $a$ is mislabeled as $b$ more than 50% of the time, the class definitions are effectively swapped, making correction unrealistic.
>
>
>
> > Later, (\hat y) becomes part of an SDE and is diffused in reverse time (Sec. 3.1.2). But now (\hat y_t) is treated almost like a continuous signal you can inject noise into, including for discrete labels. …
>
> Throughout our paper, we embed the condition inputs as continuous embeddings, regardless of whether the raw format is continuous or discrete. This is a common practice in diffusion-based conditional generation.
>
>
>
>
> > The RDC SDE in Eq. (8) is introduced with forward / reverse SDEs copied from diffusion literature, but the derivation is not convincing. …
>
> RDC is designed for pseudo-condition augmentation in early-stage training, rather than for building a dual “condition + demonstration’’ diffusion model. To make this clear, we analyze in Sec. 3.1.2 and Appendix 4.3 that RDC is mathematically equivalent to adding a time-annealed regularizer that penalizes the sensitivity of the conditional log-likelihood to condition noise, thereby enhancing the learning of pseudo-conditions.
>
>
> **2. Method description is incomplete / borderline non-reproducible**
> Several critical training details are missing or contradictory:
> > How exactly is (\hat y) updated?
>
> Empirical studies on the memorization effect show that DNNs first learn simple, clean samples and only later fit harder or noisy ones, so early stopping is applied to prevent overfitting. The parameter $\alpha$ (identical to the previously mentioned $\lambda$) and the early stopping point are used to stabilize pseudo-condition updates, with selection details in the implementation section (P22L1126, P23L1207, P23L1219). The $\lambda$ mention in Algorithm 1 is an oversight from earlier drafts, where $\lambda$ denoted the temporal ensembling momentum before being unified as $\alpha$.
>
>
> > When is the condition encoder trained and frozen?
>
>
> Throughout the paper, only the condition prediction head is frozen (as shown in Fig. 1(a)) after early stopping. There is no separate ablation exists for the two-phase procedure, as it is an integral part of RDC. In our approach, pseudo-conditions and RDC are learned and updated before early stopping. After early stopping, the model reverts to a standard conditional diffusion model, using the learned pseudo-conditions to replace noisy conditions (in the unconditional encoder case) or as supervision for the condition encoder (in the conditional encoder case).
>
> For details on epochs, steps, and related parameters, please refer to the previous response. Validation metrics are described in Appendix 5.3.
>
>
> > Sampling / guidance
>
>
> Our method focuses on handling noisy conditions during pretraining, not on estimating clean conditions at inference.
> At evaluation time, we follow standard practice and condition the diffusion model on clean test-time inputs. The conditional signal is known a priori—for example, in class-conditional image generation, we directly choose a target class and feed both the given condition and random noise into the model to generate samples.

---

> > ### Author Response · Authors · 2025-11-27
> > **Official Comment by Author**
> >
> > ## Weaknesses:
> > **3. Weak/unclear baselines and metrics**
> > The empirical claims are not convincingly supported.
> > > Label-noise baselines.
> >
> >
> > The condition prediction head of our method allows conditional diffusion to handle noisy conditions independently, addressing a different problem from standard label-noise methods. Label-noise learning methods address the $f(x) \rightarrow y$ problem, whereas our approach handles $f(x,y) \rightarrow {\nabla_{y} \log p_t(y)}$, making them structurally incompatible. One could first train a robust classifier to correct noisy labels and then use it to process the dataset before training a diffusion model. However, this approach neither generalizes to image conditions as our method does nor allows for end-to-end training.
> >
> >
> >
> >
> >
> >
> > > Metric choice for controllability.
> >
> > We evaluate controllability on the clean CIFAR-10 test set to avoid classifier corruption or distribution shift. Top-1 accuracy uses a well-pretrained classifier achieving 92.89% on clean data (Fig. 1b). For diversity, we generate 10k images evenly across 10 classes, preventing “cheating” scenarios. Mode collapse cannot occur, as metrics like FID, IS, density, and coverage, which are detailed in Appendix 5.3, would noticeably deteriorate if it did.
> >
> > > CIFAR-100 @ 60–80% noise.
> >
> > We use the official TDSM repository (https://github.com/byeonghu-na/tdsm), so there is no our implementation. The observed advantage of our method over TDSM at high noise levels reflects known limitations of transition-matrix-based approaches: estimating the transition matrix becomes very challenging at high noise.
> >
> > We also performed sanity checks, as shown in Tables 1 and 5, where we evaluated the vanilla diffusion model at  0% noise. This results raises the question of why our method can even surpass the “theoretical upper bound” at 20%  and 40% noise. This phenomenon can be explained by the observation that introducing a small amount of condition corruption in pre-training data can actually improve the quality, diversity, and fidelity of diffusion models [3]. When the noise level is 20% or 40%, our method corrects the conditions close to (but not exactly) the clean labels, and the remaining slight noise further enhances performance.
> >
> > [3] Chen, H., Han, Y., Misra, D., Li, X., Hu, K., Zou, D., ... & Raj, B. (2024). Slight corruption in pre-training data makes better diffusion models. Advances in Neural Information Processing Systems, 37, 126149-126206.
> >
> >
> >
> >
> >
> > > Visuomotor / robotics experiments.
> >
> > Our method remains robust under challenging visual perturbations, as shown by additional experiments with 60% and 80% Gaussian noise (mentioned in the Appendix 5.6). While we have not yet performed full closed-loop evaluations under unseen distortions, we acknowledge this limitation and plan to address it in future work.
> >
> >
> > **4. Mathematical rigor of RDC**
> > > RDC is pitched as a key novelty, but right now it reads like ad hoc noise augmentation with diffusion-flavored notation…
> >
> > RDC is designed for pseudo-condition augmentation in early-stage training, rather than for building a dual “condition + demonstration’’ diffusion model. To make this clear, we analyze in Sec. 3.1.2 and Appendix 4.3 that RDC is mathematically equivalent to adding a time-annealed regularizer that penalizes the sensitivity of the conditional log-likelihood to condition noise, thereby enhancing the learning of pseudo-conditions.
> >
> > Regarding the ablation study, we focused on evaluating the effect of PC and RDC. Therefore, comparisons were made under the same noise level and other identical settings (e.g., EMA schedule, guidance scale), which we consider acceptable. Of course, performing additional experiments at 20%, 60%, and 80% noise levels would provide further evidence, and we plan to include these in future work.
> >
> >
> > ## Questions:
> > > Do we really need the REVERSE-TIME DIFFUSION CONDITION? is that possible to define a manual schedule to anneal the noise level of the \hat y. I think here the author lacks enough discussion about the intuition/motivation of the introducing of such a techinique.
> >
> > We didn’t claim that RDC is the only way to enhance pseudo-condition learning. RDC leverages the well-constructed demonstration diffusion framework to define reverse diffusion for pseudo-conditions, providing smooth and controllable augmentation. This approach proves highly effective and is simpler and more stable than manually designing noise schedules.

---

### Official Review · Reviewer_pT6p · 2025-11-01

**Soundness:** 2
**Presentation:** 2
**Contribution:** 2
**Rating:** 4
**Confidence:** 3

**Summary:**

This paper introduces a training framework for conditional diffusion models designed to handle highly noisy conditioning inputs. The method combines two key components: Pseudo Condition and Reverse-time Diffusion Condition. The Pseudo Condition is produced by a lightweight prediction head that processes the UNet encoder’s output, serving as a surrogate for the clean condition and mitigating the impact of noise in the original input. Meanwhile, the Reverse-time Diffusion Condition employs a reverse SDE to generate a denoised estimate of the input condition. By integrating PC and RDC, the proposed approach effectively approximates clean conditioning signals, leading to improved performance across tasks such as label-conditioned image generation and image-conditioned visuomotor policy learning.

**Strengths:**

The paper is clearly written and well-structured, making it easy to follow. The overall presentation is coherent, and Algorithm 1 effectively clarifies the proposed method. Most of the experiments generally support the authors’ claims, with the 2D toy experiments providing particularly intuitive and convincing evidence.

**Weaknesses:**

The main concern with this paper lies in the experimental design and the limited exploration of key design choices.

In the image-conditioned visuomotor policy generation experiments, the “noisy” images are generated using only two specific distortion types. While these distortions can be regarded as forms of noise, they do not capture more general or practically relevant noise sources—such as typical camera noise or Gaussian noise. Incorporating experiments with these more realistic noise types would strengthen the method’s robustness and practical deployability in real-world scenarios.

Moreover, the experiments are confined to the UNet architecture, leaving the generalization of the proposed approach to other diffusion model backbones—particularly Transformer-based architectures—unclear. The experiments in Section 4.2 are also limited to class-label–conditioned image generation with demonstrations. Extending the evaluation to other conditioning modalities, such as line sketches (as referenced in [1]), would further demonstrate the versatility of the proposed method.

[1] Zhang, L., Rao, A., & Agrawala, M. (2023). Adding conditional control to text-to-image diffusion models. In Proceedings of the IEEE/CVF international conference on computer vision (pp. 3836-3847).

Minor:

The caption of Table 2 does not appear to correspond correctly to the content of the table.

**Questions:**

N/A

---

> ### Author Response · Authors · 2025-11-27
> **Official Comment by Author**
>
> Thank you for your comments!
>
> ## Weaknesses:
> > The main concern with this paper lies in the experimental design and the limited exploration of key design choices.
> In the image-conditioned visuomotor policy generation experiments, the “noisy” images are generated using only two specific distortion types. While these distortions can be regarded as forms of noise, they do not capture more general or practically relevant noise sources—such as typical camera noise or Gaussian noise. Incorporating experiments with these more realistic noise types would strengthen the method’s robustness and practical deployability in real-world scenarios.
>
> To address the reviewer’s concern, we conducted additional experiments with 60% and 80% Gaussian noise in our tabletop manipulation setup. The results show that our method remains robust under this setting, demonstrating its effectiveness and practical applicability. Please refer to Appendix 5.7 for more details.
>
>
>
>
> > Moreover, the experiments are confined to the UNet architecture, leaving the generalization of the proposed approach to other diffusion model backbones—particularly Transformer-based architectures—unclear. The experiments in Section 4.2 are also limited to class-label–conditioned image generation with demonstrations. Extending the evaluation to other conditioning modalities, such as line sketches (as referenced in [1]), would further demonstrate the versatility of the proposed method.
> [1] Zhang, L., Rao, A., & Agrawala, M. (2023). Adding conditional control to text-to-image diffusion models. In Proceedings of the IEEE/CVF international conference on computer vision (pp. 3836-3847).
>
> We have added experiments with a Transformer-based diffusion backbone. These results confirm that our method are not tied to a specific network architecture. Please refer to Appendix 5.8 for more details.
>
>
>
> > Minor:
> The caption of Table 2 does not appear to correspond correctly to the content of the table.
>
>
> Sorry for the typo, and we have corrected it.

---

> > ### Author Response · Authors · 2025-11-30
> > **Official Comment by Authors**
> >
> > Here we provide an additional clarification, presenting the added experiment results directly to make the response clearer and self-contained, rather than referring to the manuscript.
> >
> > ### Experiments of More Noise Sources
> > In this part, we provide additional experiments to evaluate the robustness of our method under more realistic noise conditions. Specifically, we inject extreme (60% and 80%) Gaussian noise into the image observations. As shown in Table 1, our method maintains strong performance under extreme gaussian noise, highlighting its robustness when handling Gaussian-noise images.
> >
> > **Table 1. TAC comparison with DP on the Push-T dataset under Gaussian noise.**
> >
> > | Method | 60% Noise        | 80% Noise        |
> > |:------:|:----------------:|:----------------:|
> > |   DP   | 80.32 ± 2.66     | 79.58 ± 1.51     |
> > | **Ours** | **81.47** ± **1.36** | **81.76** ± **0.62** |
> >
> >
> > ### Experiments of Transformer Backbone
> > We further examine the generality of our method by replacing the UNet backbone with a Transformer-based architecture. This evaluates whether our method remains effective under a different diffusion model backbone. As shown in Table 1, our method consistently outperforms DP when conditioned on 60% and 80% image observation distortion, demonstrating that the improvements of our method are not tied to a specific network architecture.
> >
> > **Table 1. TAC comparison with DP on the Push-T dataset under Transformer backbone.**
> >
> > | Method | 60% Noise        | 80% Noise        |
> > |:------:|:----------------:|:----------------:|
> > | DP     | 60.45 ± 2.70     | 60.54 ± **1.94**    |
> > | **Ours** | **62.72** ± **0.87** | **62.19** ± 3.47 |
> >
> >
> > We hope this response improves clarity. Thank you again for your comments.

---

### Official Review · Reviewer_hdc3 · 2025-11-01

**Soundness:** 3
**Presentation:** 2
**Contribution:** 3
**Rating:** 4
**Confidence:** 3

**Summary:**

This paper proposes a robust learning framework that helps conditional diffusion models train more effectively and efficiently under extremely noisy conditions. Building upon the theory of the *memorization effect* [1], the paper introduces *pseudo-conditions*, which gradually replace the original noisy conditions with clean conditions generated by the model itself through a *temporal ensembling* module. It further proposes *Reverse-time Diffusion Conditioning (RDC)*, which applies a reverse-time diffusion process to the pseudo-conditions as a form of conditional augmentation, enhancing the memorization effect and stabilizing training. Extensive experiments on CIFAR-10/100 and Push-T datasets demonstrate that the proposed method outperforms existing baselines (e.g., TDSM) under high noise levels.

[1] S. Liu, et al. Early-learning regularization prevents memorization of noisy labels. NeurIPS 2020

**Strengths:**

-  The paper introduces a novel robust learning approach that optimizes the conditions during training, treating noisy conditions as denoising targets for data augmentation. The method achieves significant performance gains, especially in faster convergence, showing clear benefits during the early stages of training.
-  The proposed method is lightweight and easy to integrate into existing diffusion models, making it valuable for training on large-scale, low-quality datasets.
-  The paper is well-structured, with clear organization, well-designed experiments, and detailed implementation information.

**Weaknesses:**

*   The frequent use of early stopping reduces the generalizability of the approach, suggesting that the theoretical foundation is still incomplete. The authors are encouraged to analyze the causes of overfitting and provide theoretical explanations and improvement strategies.
*   The comparison baselines are relatively limited; including more baselines would make the results more convincing.
*   The theoretical support for RDC is insufficient. Although the ablation study shows that RDC is critical to performance, further theoretical analysis and more detailed ablations are needed to clarify its contribution.

**Questions:**

- Can this method be extended to high-dimensional conditions, such as high-resolution images? In such cases, is it easy to recover pseudo-conditions from the diffusion model?
- Could you provide more textual description of Figure 2 to help readers better understand the connection between the data and your argument? Is the mention of "Figure 2(b)" in line 229 of the text a typo?
- Could you add a textual description of temporal ensembling? Although it originates from [1], as one of the new modules in the paper, and considering its early introduction, a brief explanation of its principles would help readers with limited background better understand your work and intentions.

---

> ### Author Response · Authors · 2025-11-27
> **Official Comment by Authors**
>
> Thank you for your comments!
>
> ## Weaknesses:
> > The frequent use of early stopping reduces the generalizability of the approach, suggesting that the theoretical foundation is still incomplete. The authors are encouraged to analyze the causes of overfitting and provide theoretical explanations and improvement strategies.
>
> We apply early stopping to prevent pseudo-conditions from overfitting to noisy conditions, but this trick is not fragile. Due to the memorization effect, models first fit clean condition samples (easy) and later overfit noisy ones (hard), yielding a loss curve that drops sharply and then plateaus. As we analyzed in Appendix 5.4, we stop pseudo-condition updates once the training loss stabilizes. We will further explore more principled methods to automatically decide when pseudo-conditions should be frozen.
>
>
>
> > The comparison baselines are relatively limited; including more baselines would make the results more convincing.
>
> Our baselines already cover the strongest and most directly comparable methods for this task. We welcome suggestions of any additional candidate methods to include in the final version.
>
> > The theoretical support for RDC is insufficient. Although the ablation study shows that RDC is critical to performance, further theoretical analysis and more detailed ablations are needed to clarify its contribution.
>
>
> RDC is a pseudo-condition augmentation method: by injecting Gaussian noise of varying scales into the pseudo-conditions, RDC implicitly performs a form of data augmentation. Our derivation in the revised manuscript shows that this augmentation is mathematically equivalent to adding an explicit time-annealed regularizer, which penalizes the sensitivity of the conditional log-likelihood to condition noise.
>
>
>
>
> ## Questions:
> > Can this method be extended to high-dimensional conditions, such as high-resolution images? In such cases, is it easy to recover pseudo-conditions from the diffusion model?
>
> In principle, our method embeds conditions of any type into a fixed-size latent vector before feeding them into the diffusion model, allowing it to handle conditions of arbitrary modalities and sizes. Extending the method to high-dimensional conditions, such as high-resolution images, is more challenging because recovering accurate high-dimensional pseudo-conditions is difficult due to the increased data complexity. Appendix 5.5 reports image-to-image experiments, which show limited performance and illustrate this limitation.
>
>
> > Could you provide more textual description of Figure 2 to help readers better understand the connection between the data and your argument? Is the mention of "Figure 2(b)" in line 229 of the text a typo?
>
> Inspired by Kingma et al. (2023)’s view that "diffusion models = data augmentation with increasing noise," we use diffusion to enhance pseudo-condition learning. Directly applying standard diffusion to pseudo-conditions (Figure~\ref{fig_ldc}(b)) can cause instability: at time step $t$, both the noisy demonstration $x_t$ and condition $y_t$ are nearly Gaussian, forcing the model to infer two types of information from almost structureless inputs, which may misestimate dependencies. By adjusting diffusion direction of pseudo condition, we retain partial learnable information at the boundaries, preventing training collapse.
> We apologize for the typo in the caption of Figure 2(b), which has been corrected.
>
>
> > Could you add a textual description of temporal ensembling? Although it originates from [1], as one of the new modules in the paper, and considering its early introduction, a brief explanation of its principles would help readers with limited background better understand your work and intentions.
>
> In our method, temporal ensembling is used to ensure that pseudo-conditions evolve more stably during training. As shown in Eq. 7 (P5), temporal ensembling effectively computes an exponential moving average of pseudo-conditions generated at different training steps, making the updates smoother and more stable compared to single-step predictions.

---

> > ### Author Response · Authors · 2025-11-30
> > **Official Comment by Authors**
> >
> > Here we provide an additional clarification, presenting the added experiment results directly to make the response clearer and self-contained, rather than referring to the manuscript.
> >
> > ### Experiments of Eearly Stopping Rule
> >
> > Due to the memorization effect, models first fit clean condition samples (easy) and later overfit noisy ones (hard), producing a loss curve that drops sharply before plateauing. We freeze pseudo-condition updates once the loss stabilizes. Table 1 reports visuomotor policy generation performance (TAC) under 80% camera distortion with different early stopping epochs when the loss getting stable. The results are stable across a range of epochs, demonstrating the robustness of this procedure.
> >
> > **Table 1. Visuomotor policy generation performance under 80% camera distortion.**
> >
> > | Early Stopping Epoch |  TAC            |
> > |:--------------------:|:--------------:|
> > | 80                   | 72.55 ± 1.78   |
> > | 100                  | 71.78 ± 3.24   |
> > | 120                  | 72.65 ± 1.49   |
> >
> >
> >
> > We hope this response improves clarity. Thank you again for your comments.

---

### Meta-Review · Area_Chair_ksJr · 2026-01-07

**Summary:**

The paper addresses an important problem—training conditional diffusion models under extremely noisy conditions—and proposes an interesting combination of pseudo-conditions and reverse-time diffusion. Several reviewer concerns were partially addressed in the rebuttal, including clarifications of early stopping, added experiments with additional noise types, and results on a Transformer backbone.

However, one reviewer raised substantial and detailed major concerns that remain largely unresolved. In particular, the mathematical grounding of the proposed RDC mechanism is still weak, the gap between the theoretical formulation and the actual implementation is not fully clarified, and key training details remain insufficiently specified for full reproducibility. In addition, baseline coverage is limited relative to the broader literature on learning with extreme noise, and the empirical evaluation—especially in the robotics setting—remains narrow.

Given the presence of these outstanding major concerns, especially regarding rigor and reproducibility, I do not feel confident recommending acceptance at this time, despite the practical motivation and promising empirical trends.

**Reviewer Concerns:**

Reviewer Concerns

Reviewer hdc3
The main concerns around early stopping and method clarity were addressed. The rebuttal clarified the stopping rule, added supporting experiments, and improved explanations. Concerns about theoretical depth remain, but no longer appear blocking.

Reviewer pT6p
The reviewer’s key requests were addressed. The authors added experiments with more realistic noise and a Transformer backbone, which directly resolve the generality concerns. Remaining limitations are mostly about scope.

Reviewer oTa3
Several clarifications and additional experiments were provided, but the core concerns remain. In particular, the theoretical grounding of RDC, reproducibility details, and baseline coverage are still insufficiently resolved.

Reviewer FovT
The rebuttal clarified the motivation of RDC and the early stopping procedure. However, the lack of formal analysis for RDC remains an open issue.

**Reviewer Scores:**

Reviewer hdc3

Original: 4

After discussion: 6

Rationale: Early stopping concern was clearly addressed with a concrete rule and additional results; several clarification questions were resolved. While theory remains light, the reviewer explicitly stated they “would not mind if accepted,” and the rebuttal likely removes their main doubts.

Reviewer pT6p

Original: 4

After discussion: 6

Rationale: The main concerns (noise realism and architecture generality) were directly addressed with new Gaussian-noise experiments and a Transformer backbone. These were the reviewer’s core requests.

Reviewer oTa3

Original: 2

After discussion: 2

Rationale: Although the rebuttal added experiments and clarifications, the reviewer’s major concerns on mathematical rigor, reproducibility, and baseline adequacy remain largely unresolved.

Reviewer FovT

Original: 6

After discussion: 6

Rationale: The rebuttal clarified early stopping and RDC motivation, but did not provide the formal guarantees requested. Likely no score change.

Score Summary:
Post-discussion scores: 6, 6, 2, 6
Average score: 5.0

---

### Decision · Program_Chairs · 2026-01-26

Reject